**Impacts of future land use and land cover change on mid-21[st]-**
**century surface ozone air quality: Distinguishing between the**
**biogeophysical and biogeochemical effects**
Lang Wang[1,2], Amos P. K. Tai[1,3,4], Chi-Yung Tam[1,3], Mehliyar Sadiq[1,3], Peng Wang[3],
Kevin K. W. Cheung[5]
1 Institute of Environment, Energy and Sustainability, The Chinese University of
Hong Kong, Hong Kong, China
2 Department of Geography and Resource Management, The Chinese University of
Hong Kong, Hong Kong, China
3 Earth System Science Programme, Faculty of Science, The Chinese University of
Hong Kong, Hong Kong, China
4 Partner State Key Laboratory of Agrobiotechnology, The Chinese University of
Hong Kong, Hong Kong, China
5 Department of Earth and Environmental Sciences, Macquarie University, Sydney,
Australia

24 Prepared for Atmospheric Chemistry and Physics

25 July 2020

**Abstract**

Surface ozone ($O_3$) is an important air pollutant and greenhouse gas. Land use and land cover is one of the critical factors influencing ozone, in addition to anthropogenic emissions and climate. Land use and land cover change (LULCC) can on the one hand affect ozone "biogeochemically", i.e., via dry deposition and biogenic emissions of volatile organic compounds (VOCs). LULCC can on the other hand alter regional- to large-scale climate through modifying albedo and evapotranspiration, which can lead to changes in surface temperature, hydrometeorology and atmospheric circulation that can ultimately impact ozone "biogeophysically" over local and remote areas. Such biogeophysical effects of LULCC on ozone are largely understudied. This study investigates the individual and combined biogeophysical and biogeochemical effects of LULCC on ozone, and explicitly examines the critical pathway for how LULCC impacts ozone pollution. A global coupled atmosphere-chemistry-land model is driven by projected LULCC from the present day (2000) to future (2050) under RCP4.5 and RCP8.5 scenarios, focusing on the boreal summer. Results reveal that when considering biogeochemical effects only, surface ozone is predicted to have slight changes by up to 2 ppbv maximum in some areas due to LULCC. It is primarily driven by changes in isoprene emission and dry deposition counteracting each other in shaping ozone. In contrast, when considering the combined effect of LULCC, ozone is more substantially altered by up to 5 ppbv over several regions in North America and Europe under RCP4.5, reflecting the importance of biogeophysical effects on ozone changes. In boreal and temperate mixed forests with intensive reforestation, enhanced net radiation and sensible heat induce a cascade of hydrometeorological feedbacks that generate warmer and drier conditions favorable for higher ozone levels. In contrast, reforestation in subtropical

broadleaf forests has minimal impacts on boundary-layer meteorology and ozone air
quality. Furthermore, significant ozone changes are also found in regions with only
modest LULCC, which can only be explained by "remote" biogeophysical effects. A
likely mechanism is that reforestation induces a circulation response, leading to
reduced moisture transport and ultimately warmer and drier conditions in the
surrounding regions with limited LULCC. We conclude that the biogeophysical
effects of LULCC are important pathways through which LULCC influences ozone
air quality both locally and in remote regions even without significant LULCC.
Overlooking the effects of hydrometeorological changes on ozone air quality may
cause underestimation of the impacts of LULCC on ozone pollution.

**Keywords**: ozone pollution; land use and land cover change; biogeochemical effects;
biogeophysical effects; hydrometeorology

## 1. Introduction

Surface ozone ($O_3$), as a harmful air pollutant, has negative consequences for human health (WHO, 2005; Jerrett et al., 2009; Malley et al., 2017), decreases plant gross primary productivity (e.g., Yue and Unger 2014), and leads to substantial reductions in global crop yields (Avnery et al., 2011; Tai et al., 2014; Tian et al., 2016; Tai and Val Martin, 2017; Mills et al., 2018). It is also an important greenhouse gas, contributing to climate change (Myhre et al., 2013). Surface ozone is produced by the photooxidation of precursors including carbon monoxide (CO), methane ($CH_4$), and other non-methane volatile organic compounds (NMVOCs) in the presence of nitrogen oxides ($NO_x$). These precursors are both generated by human activities and naturally emitted from vegetation and soils. The dominant sink of surface ozone is photochemical loss and dry deposition to the surface including vegetation mainly in the form of leaf stomatal uptake. Depending on all of these production and loss mechanisms, its concentration is highly sensitive to changes in natural and anthropogenic emissions of precursors (Wang et al., 2011), land use and land cover (Ganzeveld et al., 2010; Val Martin et al., 2015; Fu and Tai, 2015) and climate (Jacob and Winner, 2009; Fiore et al., 2012; Schnell et al., 2016). Recent studies found that decreases in anthropogenic emissions alone might not necessarily decrease ozone in some polluted regions if factors such as climatic and land cover changes act to enhance ozone and offset emission control efforts (Zhou et al., 2013; Zhang et al., 2014; Xue et al., 2014).

Land use and land cover change (LULCC) can modify ozone concentration by altering key drivers of ozone such as biogenic VOC emissions and dry deposition (e.g., Wong et al., 2018). These can be referred to as "biogeochemical effects" of LULCC on ozone (as opposed to "biogeophysical effects", which will be discussed

next), because these processes entail directly modifying the biosphere-atmosphere
exchange of gases and particles that alters atmospheric composition including ozone
itself. Here we limit the "biogeochemical effects" of LULCC on ozone to processes
that influence ozone directly in a given climate, including biogenic VOC emission
and the dry deposition of ozone and its precursors; climatic changes that can arise
from land cover disturbances of the biogeochemical cycles are not the focus.

LULCC can modify the spatial pattern and magnitude of isoprene emission

due to their strong dependence on vegetation type and leaf density (Guenther et al.,
2012). For instance, Lathière et al. (2006) found as much as a 29% decrease in global
isoprene emission from a scenario in which 50% tropical trees are replaced by
grasses. Heald and Spracklen (2015) estimated the net effect of LULCC under future
anthropogenic influences as a decrease of 12–15% in annual isoprene emission
globally. These changes in isoprene emission can in turn modify ozone concentration.
For example, Tai et al. (2013) found that LULCC projections in the
Intergovernmental Panel on Climate Change (IPCC) A1B scenario with widespread
crop expansion could reduce isoprene emission by ~10% globally compared with the
land use and land cover at present. Such a reduction could correspondingly lead to an
up to 4 ppbv of ozone decrease in the eastern US and western Europe, and an up to 6
ppbv increase in South and Southeast Asia, whereby the difference in the sign of
responses is driven primarily by the different ozone production regimes.

Dry deposition is another key factor modulating ozone (e.g., Wesely, 1989;

Val Martin et al., 2014; Lin et al., 2019). Dry deposition is the most efficient over
densely vegetated regions via the stomatal uptake of ozone and its precursors, and
LULCC can alter these fluxes. Kroeger et al. (2013) found that reforestation over
peri-urban areas in Texas, USA, could effectively enhance dry deposition, resulting in
decreases in ozone and its precursors. Fu and Tai (2015) found that LULCC driven by
climate and $CO_2$ changes could overall enhance dry deposition and decrease ozone by
up to 4 ppbv in East Asia during the past three decades. The dry deposition
enhancement mostly arises from climate- and $CO_2$-induced increase in leaf area index
(LAI), which more than offsets the compensating effect of cropland expansion (Fu
and Tai, 2015). The relative importance of isoprene emission and dry deposition,
which could have counteracting effects on ozone given the same LULCC, is strongly
dependent on local $NO_x$ concentrations and vegetation type (Wong et al., 2018).

LULCC can also affect weather and climate by perturbing the biosphere-

atmosphere exchange of water and energy fluxes (e.g., Betts, 2001; Bonan, 2016;
Pitman et al., 2009). For example, afforestation generally cools the surface in tropical
regions, where evaporative cooling generally exceeds radiative warming from
reduced albedo, but warms the surface in boreal forests due to the more dominant
radiative warming effect (e.g., Arora and Montenegro, 2011; Lee et al., 2011; Bonan,
2008). There is little consensus on the effects of afforestation in midlatitude regions
(e.g., Boisier et al., 2012; de Noblet-Ducoudré et al., 2012). Recent studies (Devaraju
et al. 2015; Laguë and Swann 2016) have identified that LULCC in midlatitude
regions can modify the global energy balance, impacting cloud cover, precipitation,
and circulation pattern. Furthermore, the impacts of such surface forcing could extend
into the upper troposphere, alter large-scale circulation pattern, and consequently
affect the climate in remote regions (Henderson-Sellers et al. 1993; Chase et al., 2000;
Swann et al., 2012; Medvigy et al., 2013). Laguë et al. (2019) examined the climatic
effects of individual physical components in the land surface (albedo, evaporative
resistance and surface roughness), and found that temperature responds most to
changes in albedo and evaporative resistance through large-scale atmospheric
feedbacks. Still, how individual land characteristics play out together and interact
with each other to affect the atmospheric general circulation are not fully understood.

By and large, the impacts of LULCC on weather and climate are complex.

There is high confidence that LULCC can affect regional climate and climate in
remote areas as far as few hundreds of kilometers away (Jia, et al., 2019). The
magnitude and sign of regional climate change vary across regions depending on the
magnitude of LULCC and background climatic conditions. However, on the global
scale, the net changes resulting from LULCC alone are relatively small (e.g.,
Matthews et al. 2004; Pongratz et al. 2010; Brovkin et al., 2013; Shevliakova et al.
2013; Simmons and Matthews, 2016). Thus, sometimes climatic responses to LULCC
may be difficult to distinguish from natural climate variability especially on the global
scale.

The modification of the overlying meteorological environment and climate

induced by LULCC and the associated exchange of momentum, heat and moisture
between the land and atmosphere can be defined as "biogeophysical effects" of
LULCC. Such effects can further alter surface ozone on local to pan-regional scales
(Jiang et al., 2008; Ganzeveld et al., 2010; Wu et al., 2012), and we shall call these
and related pathways the biogeophysical effects of LULCC on ozone. In particular, a
LULCC-induced increase in surface temperature could (1) accelerate peroxyacetyl
nitrate (PAN) decomposition into $NO_x$ (Jacob and Winner, 2009; Doherty et al., 2013;
Pusede et al., 2015), (2) increase biogenic VOCs emissions from vegetation
(Guenther et al., 2012; Wang et al., 2013; Squire et al., 2014), and (3) lead to more
water vapor in air that tends to increase ozone destruction (Jacob and Winner, 2009).
The net effect of higher temperatures is almost always ubiquitously an enhancement
of ozone levels reported from both observational (e.g., Porter et al., 2015; Pusede et
al., 2015) and modeling (e.g., Shen et al., 2016; Lin et al., 2017) studies in many
polluted regions. Meanwhile, any reduction in precipitation, cloud cover and soil
moisture can also enhance surface ozone because of the associated increase in solar
radiation and reduced dry deposition velocity. Fig. 1 summarizes the possible
biogeochemical and biogeophysical pathways through which a change in forest
coverage may influence surface ozone. The relative importance of different pathways,
many of which may either counteract or amplify each other, is strongly dependent on
forest types.

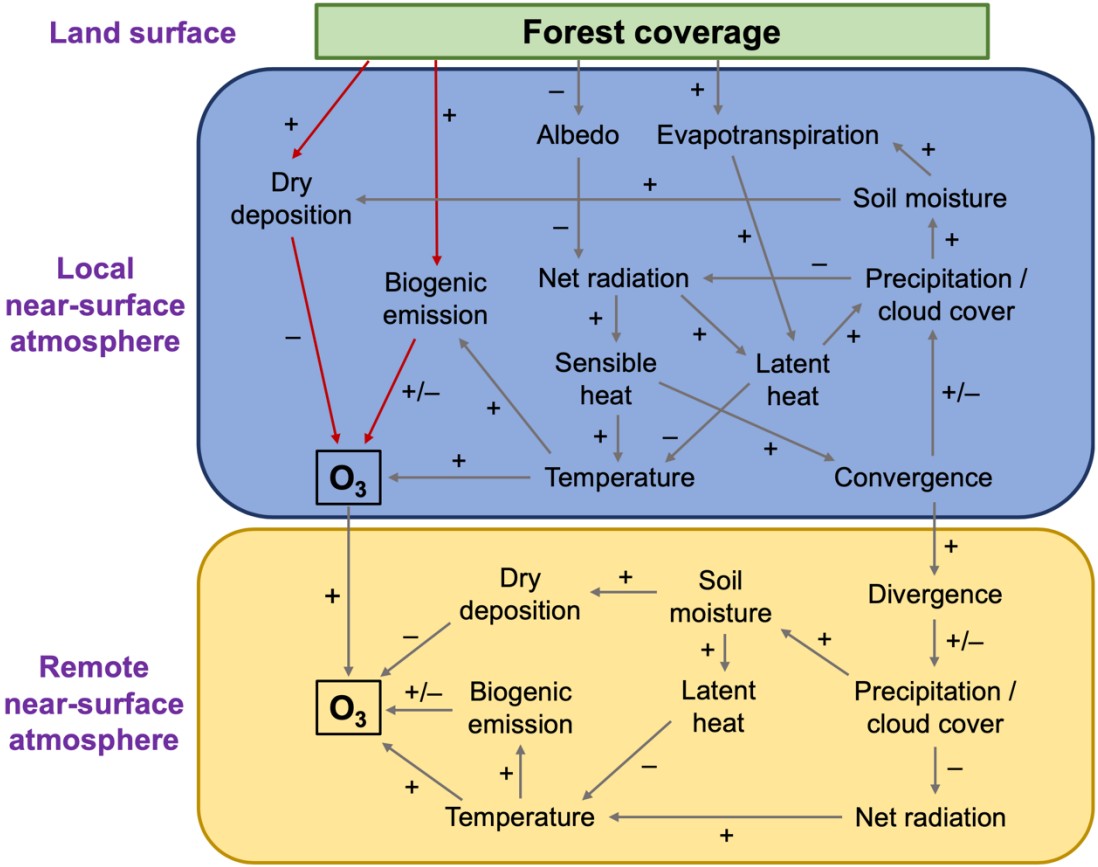


Figure 1. Schematic diagram showing the biogeochemical and biogeophysical effects of any changes in
the forest cover resulting from land use and land cover change (LULCC) on surface ozone. Red arrows
indicate the biogeochemical pathways and grey arrows indicate the biogeophysical effects via changes
in the overlying meteorological environment. The sign associated with each arrow indicates the
correlation between the two variables; the sign of the overall effect (positive or negative) of a given
pathway is the product of all the signs along the pathway. We here focus on processes initiated on the
land surface by LULCC, and the corresponding responses in local near-surface atmosphere (blue box)
and remote near-surface atmosphere (yellow box).

The LULCC biogeophysical effects have thus far been largely unexplored,

though biogeochemical effects of LULCC have been examined by a number of
studies (Wu et al., 2012; Fu and Tai, 2015; Heald and Geddes, 2016). Only a few
recent studies have implicitly included such biogeophysical effects of LULCC in their
coupled land-atmosphere models when assessing the impacts of LULCC on surface
ozone. Val Martin et al (2015) studied the combined effects of LULCC on surface
ozone using future LULCC scenarios, and found an increase of 2–3 ppbv from 2000
to 2050 over US national parks. Ganzeveld et al. (2010) also calculated the future
LULCC from 2000 to 2050, and found that an increase in boundary-layer ozone
mixing ratios by up to 20% over the tropics. However, these studies did not
distinguish between the roles of biogeophysical vs. biogeochemical effects, or
decipher the physics and relative importance of various mechanisms behind the
combined effects.

The aim of this study is to investigate how and to what extent global LULCC

could affect surface ozone in the near future by investigating and distinguishing
between the biogeochemical, biogeophysical and combined effects of LULCC. We
suggest a new line of biogeophysical pathways linking LULCC to surface ozone, and
also consider biogeochemical pathways through isoprene emission and dry deposition
changes caused by LULCC. In particular, over the regions without significant
LULCC but showing substantial ozone changes, we find that the biogeophysical
effects arising from LULCC-induced atmospheric circulation changes can be
dominant and could be isolated from the combined effects. LULCC is one of the key

strategies for climate change mitigation, but meanwhile has substantial impacts on

ozone pollution. Understanding its comprehensive pathways on surface ozone can

help provide important references for integrated air quality and land use management

in the future.

**2. Data and methods**

*2.1 Modeling framework*

To simulate the impacts of LULCC on surface ozone, we use the Community

Earth System Model (CESM) version 1.2 (http://www.cesm.ucar.edu/models/), which

is a comprehensive global model that couples different independent components for

the atmosphere, land, ocean, sea ice, land ice and river runoff (Lamarque et al., 2012).

The atmospheric component is the Community Atmosphere Model version 4

(CAM4), which uses a finite-volume dynamical core with comprehensive

tropospheric and stratospheric chemistry (CAM-Chem). Chemical mechanisms are

based on the Model for Ozone and Related chemical Tracers (MOZART) version 4

(Emmons et al., 2010). For the land component, the Community Land Model (CLM)

version 4.5 (Oleson, 2013) considers 16 Plant Function types (PFTs) (Lawrence et al.,

2011), and prescribes the total leaf area index (LAI), the PFT distribution and PFT-

specific seasonal LAI derived from Moderate Resolution Imaging Spectroradiometer

(MODIS) observations. We use the Satellite Phenology (SP) mode of CLM4.5 for all

simulations, which prescribes vegetation structural variables including LAI and

canopy height; active biogeochemical cycling in terrestrial ecosystems is not turned

on.

In CLM4.5, biogenic VOC emissions are computed using the Model of

Emissions of Gases and Aerosols from Nature (MEGAN) version 2.1 (Guenther et al.,

2012), accounting for the major known processes controlling biogenic VOC

emissions from terrestrial ecosystems, such as effects of temperature, solar radiation,
soil moisture, leaf age, $CO_2$ concentrations, and vegetation species and density.
Biogenic VOC emissions in MEGAN are allowed to respond interactively to changes
of these processes. Thus, isoprene emission is allowed to respond to spatiotemporal
changes in PFTs and the associated changes in meteorological conditions in this
study. Dry deposition of gases and aerosols are computed based on the multiple
resistance approach of Wesely (1989), updated by Emmons et al. (2010), Lamarque et
al. (2012) and Val Martin et al. (2014). In the scheme, dry deposition velocity is the
inverse of aerodynamic resistance ($R_a$), sublayer resistance ($R_b$) and bulk surface
resistance ($R_c$), whereby $R_c$ includes a combination of resistances from vegetation
(including stomatal resistance), lower canopy, and ground with specific values for
different land types. Correspondingly, dry deposition velocity in the scheme responds
to primarily meteorological and ecophysiological conditions. Soil $NO_x$ emissions are
dependent on soil moisture, soil temperature and vegetation cover (Emmons et al.,
2010; Yienger and Levy, 1995), while biomass burning emissions and anthropogenic
emissions of ozone precursors, are prescribed by inventory at present-day levels.

The coupled CAM-Chem-CLM model configuration of CESM can be run

with prescribed meteorology to drive atmospheric chemistry-only simulations
(hereafter as dynamical Off-line mode), or with interactive, dynamically simulated
meteorology using CAM4 (hereafter as On-line mode). These two modes are both
applied in the study. In particular, the Off-line mode is used to quantify the
biogeochemical effects of LULCC alone on surface ozone in the absence of any
associated meteorological responses to LULCC. The On-line mode is applied to
assess the biogeophysical and combined effects on ozone caused by LULCC,
considering also the effects of the resulting meteorological changes.

256   For the Off-line mode, we use the Goddard Earth Observing System Model

257 Version 5 (GEOS-5) (https://rda.ucar.edu/datasets/ds313.0/) (Tilmes, 2016)

258 assimilated meteorology as the driving fields, with a horizontal resolution of

259 $1.9° \times 2.5°$ and 56 vertical levels between the surface and the 4-hPa level. For the On-

260 line mode of CAM4-Chem-CLM, 26 vertical levels are used between the surface and

261 4 hPa, with the same horizontal resolution as the Off-line mode. For all simulations,

262 concentrations of long-lived greenhouse gases including $CO_2$, $CH_4$, and $N_2O$ are

263 prescribed at present-day. For the anthropogenic emissions used for all simulation are

264 described in Lamarque et al. (2010, 2012) and references therein. Climatic changes

265 that may arise from land cover disturbances of the terrestrial carbon and nitrogen

266 cycles are not the focus of this study, which aims to delineate the more immediate

267 responses of surface ozone to LULCC.

268   The CAM-Chem-simulated atmospheric chemistry has been extensively

269 evaluated and documented (e.g., Lamarque et al., 2012). In general, CAM-Chem can

270 reasonably replicate observed values at individual sites (CASTNET for US and

271 EMEP for Europe) (Lamarque et al., 2012; Val Martin et al., 2014; Sadiq et al.,

272 2017), and mid- and upper-tropospheric distribution derived from a compilation of

273 ozone measurements (Lamarque et al., 2010; Cooper et al., 2010), albeit with a

274 general overestimation. The performance is comparable to other global and regional

275 models (Lapina et al., 2014; Parrish et al., 2014). Uncertain emissions, coarse

276 resolution (Lamarque et al., 2012), misrepresentation of dry deposition process (Val

277 Martin et al., 2014) and overestimation of stomatal resistance (Lin et al., 2019) are all

278 likely factors contributing to the biases.


 *2.2 Present and future land use and land cover scenarios*

For the present-day land cover distribution, satellite phenology based on

MODIS and a cropping dataset from Ramankutty et al. (2008) are used (see Lawrence
et al., 2011). The cropping dataset combines agricultural inventory data and two
satellite-derived land products. For the future land cover, projections based on the
Representative Concentration Pathways (RCP) 4.5 and 8.5 scenarios are adopted (van
Vuuren et al., 2011). Both are computed using Integrated Assessment Models (IAM)
for the Phase 5 of the Coupled Model Intercomparison Project (CMIP5) community,
incorporating anthropogenic transformation and activities associated with carbon
releases (e.g., wood harvest). These LULCC projections are internally consistent with
the corresponding emission scenarios and development pathways for the Fifth
Assessment Report (AR5) of Intergovernmental Panel on Climate Change (IPCC)
(Taylor et al., 2012). In general, the RCP4.5 LULCC has the most extensive use of
land management as a carbon mitigation strategy, with the expansion of forest areas
combined with large reductions in croplands and grasslands. The RCP8.5 LULCC has
the least effective use of land management for carbon mitigation, with large
expansion in both croplands and grasslands together with substantial forest losses. In
this study, anthropogenic emissions are held constant at the present-day level for all
runs, thus the effects of LULCC can be considered as being decoupled from changes
in anthropogenic emissions in order to isolate the effects of LULCC alone.

Both present-day and future land cover are transformed into PFTs changes for

implementation into CESM (Lawrence et al., 2012; Oleson et al., 2013). The long-
term time series of LULCC span through the historical (1850–2005) and future
(2006–2100) periods in 5-year intervals (Riahi et al., 2007; van Vuuren et al., 2007;
Wise et al., 2009a), and are then interpolated and harmonized with smooth transitions
on the annual timescale (Hurtt et al., 2011). For this work, we focus on LULCC from
the present-day (2000) to future (2050) period.

*2.3 Model experiments*

We have two sets of configuration, Off-line mode and On-line mode to

investigate the impacts of LULCC on surface ozone (see Table 1). We focus on boreal
summer month (June-July-August, JJA) averages as this is the period when ozone
pollution is generally the most severe in the Northern Hemisphere. In the first set of
simulations in Off-line mode, surface ozone would respond to LULCC only through
biogeochemical effects that mainly include changes in dry deposition velocity and
isoprene emissions without meteorological responses to LULCC. The Off-line mode
includes control run (Off-line_CTL) using present-day (year 2000) distribution of
land use and land cover, and two future simulations Off-line_45 and Off-line_85, with
year-2050 land use and land cover distribution following RCP4.5 and RCP8.5,
respectively. All three experiments are time-sliced simulations using prescribed
GEOS-5 meteorology from 2004 to 2017 for 14 years allowing for interannual
climate variability, and we use the last 10-year averages for analysis. The statistical
significance of the comparison amongst these experiments was assessed by the
Student's t-test at the 95% confidence levels.

| | Case Name | Land treatment | Meteorology | Simulated years | Model forcing |
|---|---|---|---|---|---|
| 1 | Off-line_CTL | Present-day (2000) land use and land cover (LULC) map | GEOS-5 reanalysis (2004-2017) | 14 years, the last 10 years for analysis | - Present-day (2000) well-mixed greenhouse gases and short-lived gases and aerosols, anthropogenic emissions; - Present-day (2000) monthly mean sea surface temperature and sea ice -All simulations use the SP mode in CLM - Isoprene emission is from MEGAN - Dry deposition velocity is based on Wesely (1989) updated by Val Martin et al. (2014) |
| 2 | Off-line_45 | 2050 RCP4.5 future LULC map as a time slice | Same as above | Same as above | |
| 3 | Off-line_85 | 2050 RCP8.5 future LULC map as a time slice | Same as above | Same as above | |
| 4 | On-line_CTL | Present-day (2000) LULC map | Simulated online | 60 years (looped over same year of forcing), the last 30 years for analysis | |
| 5 | On-line_45TS | 2050 RCP4.5 future LULC map as a time slice | Same as above | Same as above | |
| 6 | On-line_85TS | 2050 RCP8.5 future LULC map as a time slice | Same as above | Same as above | |
| 7, 8 | On-line_45[a] | 2000-2005 historical, 2006-2065 RCP4.5 transient LULC map | Same as above | 66 years (transient land forcing all the way), the last 30 years[c] for analysis | |
| 9, 10 | On-line_85[b] | 2000-2005 historical, 2006-2065 RCP8.5 transient LULC map | Same as above | Same as above | |


Table 1. List of model experiments. [a, b] Case 8 and 10 are in On-line_45 and On-line_85 are similar to

Case 7 and 9, respectively, but with slightly different initial conditions to produce two ensemble

members. [c] The analysis time period is from 2036 to 2065, centered around year 2050, as part of the
transient land forcing.

In the second set of On-line mode simulations, ozone would respond to both
the biogeochemical and biogeophysical effects caused by future projected LULCC.
The first experiment On-line_CTL, reflects present-day conditions and uses land
surface forcing for year 2000. The second and third experiments, On-line_45TS and
On-line_85TS, are time-sliced simulations using 2050 land cover distribution
following RCP4.5 and RCP8.5, respectively. These two experiments are designed for
direct, parallel comparison with the Off-line simulations, except with longer
integration (60 years) and analysis (30 years) time to capture interannual climate
variability. Because these multi-year simulations are looped over the same year of
land cover forcing, they can be considered as a quasi-ensemble run and the multi-year
average can be considered as the ensemble average. The fourth and fifth experiments,
referred to as On-line_45 and On-line_85, are transient simulations performed
continuously from year 2000 to 2065 using transient land cover maps projected for
the RCP4.5 and RCP8.5 scenarios, respectively. These On-line transient simulations
are repeated by a series of ensemble runs with slightly different initial conditions,
with two ensemble members for each scenario. All the On-line experiments analysis
is based on the last 30-year average and the ensemble average when modeled
variables have attained a quasi-steady state. Comparison between the time-sliced and
transient simulations helps us ascertain the strengths of LULCC-induced climate
signals.
All simulations are performed with prescribed sea surface temperature and
sea-ice cover following the HadISST data set (Rayner et al., 2003) at the year-2000
level. Long-lived greenhouse gases and thus the radiative forcing from them are kept
at present-day conditions (year 2000) to isolate the effects of LULCC only.

These model configurations allow us to separate and examine: (1)

biogeochemical effects of LULCC on surface ozone, (2) biogeophysical effects on
surface ozone, and (3) the combined effects induced by LULCC on surface ozone and
its precursors and dry deposition.

**3. Results**
*3.1 Projected land use and land cover change from 2000 to 2050*

Figure 2 shows the global distribution of present-day (year 2000) PFTs and

future projected changes (2000 to 2050) following RCP4.5 and RCP8.5 for three
major land cover categories. The future LULCC in RCP4.5 is characterized by
extensive forest expansion (Figs. 2f, g). Transition from present-day to 2050 in
RCP4.5 highlights the global growth of forest from 71.8 million to 74.0 million $km^2$,
at the expense of croplands (from 14.7 million to 12.3 million $km^2$); grasslands
slightly increase in area from 33.7 million to 33.8 million $km^2$. The net increase of 2.2
million $km^2$ of forests is consistent with that provided by Hurtt et al. (2011),
Lawrence et al. (2012) and Heald and Geddes (2016). Fig. 2f also illustrates cropland
area increases over Southeast Asia, India and China. Such increases are due to more
bioenergy crop production for the purpose of climate change mitigation, economic
advantages from agriculture productivity growth, lower regional land prices, and
availability of undeveloped lands in these developing regions (Wise et al., 2009b;
Thomson et al., 2011). In contrast, regions such as Europe, US and Canada, undergo
extensive reforestation. RCP8.5 LULCC is characterized by extensive cropland
expansion (Figs. 2k, l, m), driven mainly by a large increase in the global population
and a slow increase in crop yields due to a slow rate of exchange of technology
globally (Riahi et al., 2011). Cropland expansion occurs largely over the tropical belt
(30°N-30°S) at the expense of forest reduction. The total increases in croplands are by
1.8 million km$^2$, and forest area decreases by 2.5 million km$^2$.

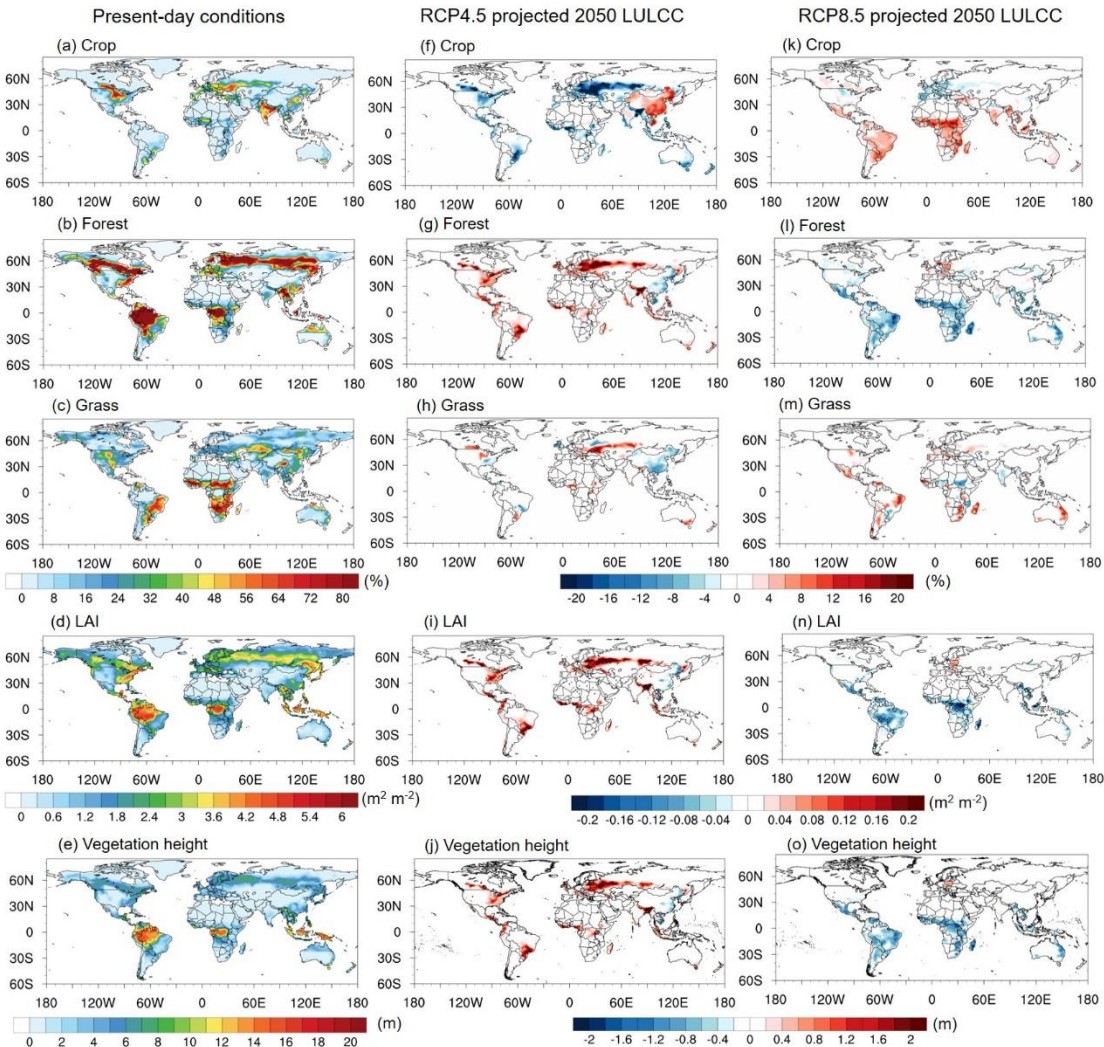


Figure 2. Present-day (2000) land use and land cover by percentage of land coverage, total leaf area
index (LAI) and vegetation height (left), and their changes from 2000 to 2050 under RCP4.5 (middle)
and RCP8.5 (right) scenarios for the boreal summer (June-July-August) (units at the right side of the
color bar). Plant function types (PFTs) in CESM are here grouped into three major categories: crop,
forest and grass. The treatment of vegetation including PFT fractional coverage, LAI and vegetation
height is prescribed using the SP mode of CLM4.5 in both the present-day case and future LULCC
scenarios. For the future cases, PFT fractional coverage is derived according to the RCP land scenarios.

The present-day LAI and its changes associated with the future projected

LULCC are shown in Figs. 2d, 2i and 2n. Forest expansion leads to increases in LAI,
whereas deforestation results in LAI reduction. For RCP4.5, due to the widespread
reforestation and afforestation except in East Asia, LAI increases significantly.
Particularly over Europe and the US, the absolute increase in LAI is > 0.1. For
RCP8.5, LAI generally declines with intense reductions over the tropical regions.

*3.2 Biogeochemical effects of land use and land cover change on surface ozone*

Figure 3 shows the simulated changes in ozone concentrations, isoprene

emission rates and dry deposition velocities based on the Off-line simulations. We
find that isoprene emission changes correspond closely with the LULCC in each
future scenario from 2000 to 2050 (Figs. 3b, e). For RCP4.5, isoprene emission
increases over the regions with forest expansion, including the US, Europe and some
tropical regions, but decreases over East Asia. Such isoprene emission increases are
primarily driven by forest expansion, since forest PFTs typically emit much more
isoprene than crops and grasses (Guenther et al., 2012). For RCP8.5, isoprene
emission decreases over the tropics with slight increases over Europe, north China
and north India, largely due to forest reduction in this scenario.

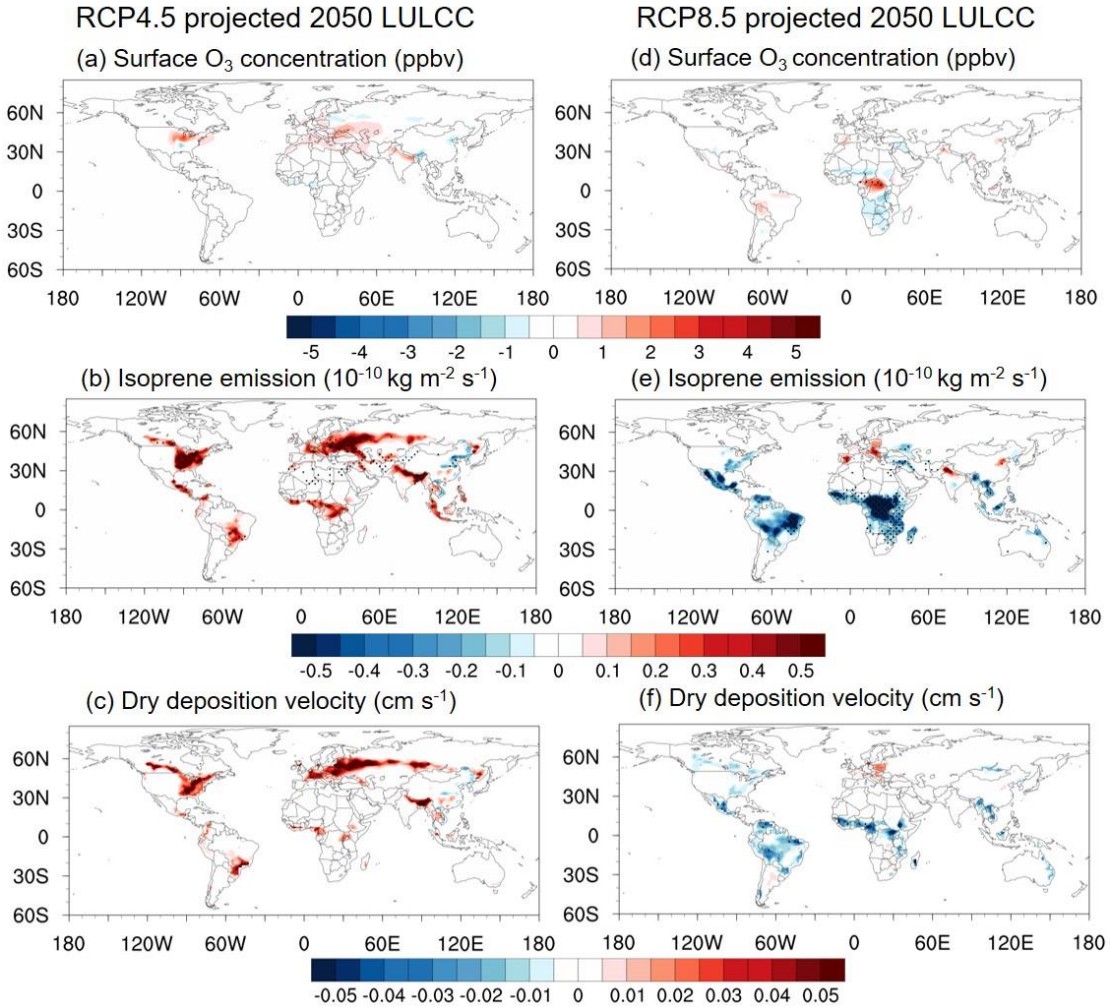


Figure 3. Simulated 2000-to-2050 changes in surface ozone, isoprene emission, and dry deposition

velocity under RCP4.5 and RCP8.5 projected LULCC for the boreal summer (June-July-August)

averaged for the final 10 years of simulations. Regions with dots indicate changes that are significant at

the 95% confidence level. These are results from Off-line runs with prescribed meteorology; i.e.,

meteorological variables do not respond to LULCC.

Table 2 summarizes the percentage and absolute changes of the annual global

isoprene emission. The simulated present-day annual global isoprene is 353.8 Tg C

$yr^{-1}$, in the middle of the range 308–678 Tg C $yr^{-1}$ summarized by Guenther et al.

(2012). For the RCP4.5 LULCC, the annual global isoprene emission increases by

5.2%, but it decreases by 11.8% for RCP8.5. The isoprene emission changes are in

line with these studies by Heald et al. (2008) and Wu et al. (2012), who estimated a

decrease of 12–15% in global isoprene emission under the net biogeochemical effect
of future LULCC (A1B and A2 scenarios).

| | | Isoprene emissions (TgC yr$^{-1}$) | % change | Ozone dry depositional sink (Tg yr$^{-1}$) | % change | Ozone concentration (ppbv) | % change |
|---|---|---|---|---|---|---|---|
| Off-line | Off-line_CTL | 353.8 | | 886.8 | | 23.6 | |
| | Off-line_45 | 372.3 | 5.2 | 895.4 | 1.0 | 23.7 | 0.4 |
| | Off-line_85 | 311.9 | –11.8 | 879.8 | –0.8 | 23.5 | –0.4 |
| On-line | On-line_CTL | 417.7 | | 969.2 | | 26.2 | |
| | On-line_45TS | 435.4 | 4.3 | 974.7 | 0.6 | 26.5 | 1.2 |
| | On-line_85TS | 386.8 | –7.4 | 964.1 | –0.5 | 26.4 | 0.8 |
| | On-line_45 | 440.3 | 5.5 | 975.6 | 0.6 | 26.6 | 1.5 |
| | On-line_85 | 385.2 | –7.7 | 964.1 | –0.5 | 26.3 | 0.4 |

Table 2. Annual average global isoprene emission and ozone dry-depositional sink as influenced by
future LULCC in the RCP4.5 and RCP8.5 scenarios; shown separately are changes in prescribed
meteorology (biogeochemical effects only) and coupled atmosphere-chemistry-land configurations
(both biogeochemical and biogeophysical effects).

Fig. 3c shows that LULCC in the RCP4.5 scenario has enhanced dry

deposition velocity over most regions where forests have expanded. Forest with both
large LAI, and high surface roughness often provides the highest dry deposition
velocity amongst all PFTs (Emmons et al., 2010; Lamarque et al., 2012). The most
dramatic changes occur in Europe where local maximum changes occur in land cover
between forests and croplands. Local decreases over East Asia are the result of
deforestation. For RCP8.5, dry deposition velocity decreases mostly over the regions
where tropical forests are replaced by croplands (Fig. 3f). Equatorial Africa and the
Amazon experience the largest decrease in dry deposition velocity relative to present-
day conditions. Some increases over Western Europe are the result of local
reforestation.

The globally averaged change in the dry-depositional sink is around 1%

(Table 2). Local dry deposition velocity changes within 0.05 cm s$^{-1}$. The value of dry
deposition velocity change is in line with previous studies exploring future 2050
LULCC alone on the dry deposition velocity of ozone (e.g., Verbeke et al., 2015),
though our results show slightly larger changes due to larger LAI differences between
forests and crops/grasses during the boreal summer compared with their annual mean
values of differences from Verbeke et al. (2015).

Figs. 3a and 3d show the impacts of future projected LULCC on surface

ozone. LULCC under RCP4.5 with massive forest expansion increases isoprene
emission that could increase surface ozone, but also enhance dry deposition velocity
that could reduce surface ozone. The overall changes in surface ozone are thus
generally small due to these compensating effects. There are a few regions with
surface ozone changes by up to 2 ppbv. In particular, over the US, opposite surface
ozone changes are seen in RCP4.5: an increase in the northeast US and a decrease in
the southeast US despite of the fact that both changes are driven by forest expansion
(Fig. 3a). Such a contrasting pattern is shaped by the local atmospheric chemical
conditions related to $O_3$-$NO_x$-VOC chemistry. The northeast US is a high-$NO_x$ region,
and increases in isoprene emission result in enhanced ozone, more than offsetting the
effect of increasing dry deposition velocity. In contrast, the southeast US is a high-
isoprene-emitting region; additional isoprene may react with ozone and $NO_x$, thereby
suppressing surface ozone production (Kang et al., 2003; von Kuhlmann et al., 2004;
Fiore et al., 2005; Pfister et al., 2008). Furthermore, in the low-$NO_x$ region, OH is
largely removed by reactions with biogenic VOCs, producing peroxy radicals that
form $HO_2$ or producing organic peroxides. Recent studies found that these peroxides
can be rapidly photolyzed, making them at best a temporary $HO_x$ reservoir (e.g.,
Thornton et al., 2002; Kubistin et al., 2010). This result implies that in low-$NO_x$
regions ozone production may be $NO_x$-saturated more often than current models
suggest. Suppressed ozone is also found in the tropical regions of South America and
Africa (Fig. S1a). Together with the increase in dry deposition velocity, overall there
is a decrease of surface ozone. Similar to the northeastern US conditions, southern
Europe, northeastern India and northern China are also high-$NO_x$ regions.

Under the RCP8.5 scenario with substantial cropland and grassland expansion,

decrease in isoprene emission and dry deposition again offset each other in
controlling surface ozone in high-$NO_x$ regions. Surface ozone concentration decreases
by around 1 ppbv over the north-central and southern Africa, but increases by up to 2
ppbv over equatorial Africa and central South America (Fig. 3d). In particular, the
area with enhanced ozone in these regions corresponds well with reductions in
isoprene emission and dry deposition together. Equatorial Africa is a high-isoprene-
emitting, low-$NO_x$ region, thus decreases of isoprene emission together with reduced
dry deposition would lead to enhanced ozone (Fig. S1b).

*3.3 Biogeophysical effects of land use and land cover change on surface ozone*

Next, we examine results from the On-line simulations, which allow us to

assess the impacts of LULCC on surface ozone when the overlying meteorological
environment is also modified by LULCC. Fig. 4 shows the simulated changes in
ozone concentrations, isoprene emissions rates, dry deposition velocities as well as 2-
m air temperature from the On-line time-sliced simulations. The simulated changes in
surface ozone is in the range from –2 to +5 ppbv (Figs. 4a, e). The magnitude of
ozone changes in On-line simulations is overall larger than those in Off-line
simulations (Fig. 3 and Table 2), which consider biogeochemical effects only,
indicating the importance of complications from the changing meteorological
environment in response to LULCC. Within the On-line simulations, more substantial
responses of meteorology as well as of surface ozone to LULCC are found in RCP4.5
compared with those in RCP8.5.

In contrast to the clear, localized signals in ozone changes in response to

LULCC through biogeochemical pathways, surface ozone changes are more complex
when biogeophysical pathways are also involved (Figs. 4a, e). Most importantly, both
local and remote ozone changes can be discerned. Such signals are not captured by
the Off-line simulations in which changes only respond to LULCC locally (Fig. 3).
Furthermore, changes in 2-m air temperature are found to be correlated well with
patterns of changes in ozone (Fig. S2a, d), indicating that the biogeophysical drivers
that modify meteorological conditions may play critical roles in ozone changes. Figs.
4d and 4h show simulated changes in 2-m air temperature before and after LULCC.
Regional-scale temperature changes of up to 2 K are found. Such magnitudes of
temperature anomalies induced by LULCC are in line with those from previous
experiments (Lawrence et al., 2012; Brovkin et al., 2013). Over the regions where
temperature increases, surface ozone increases correspondingly.

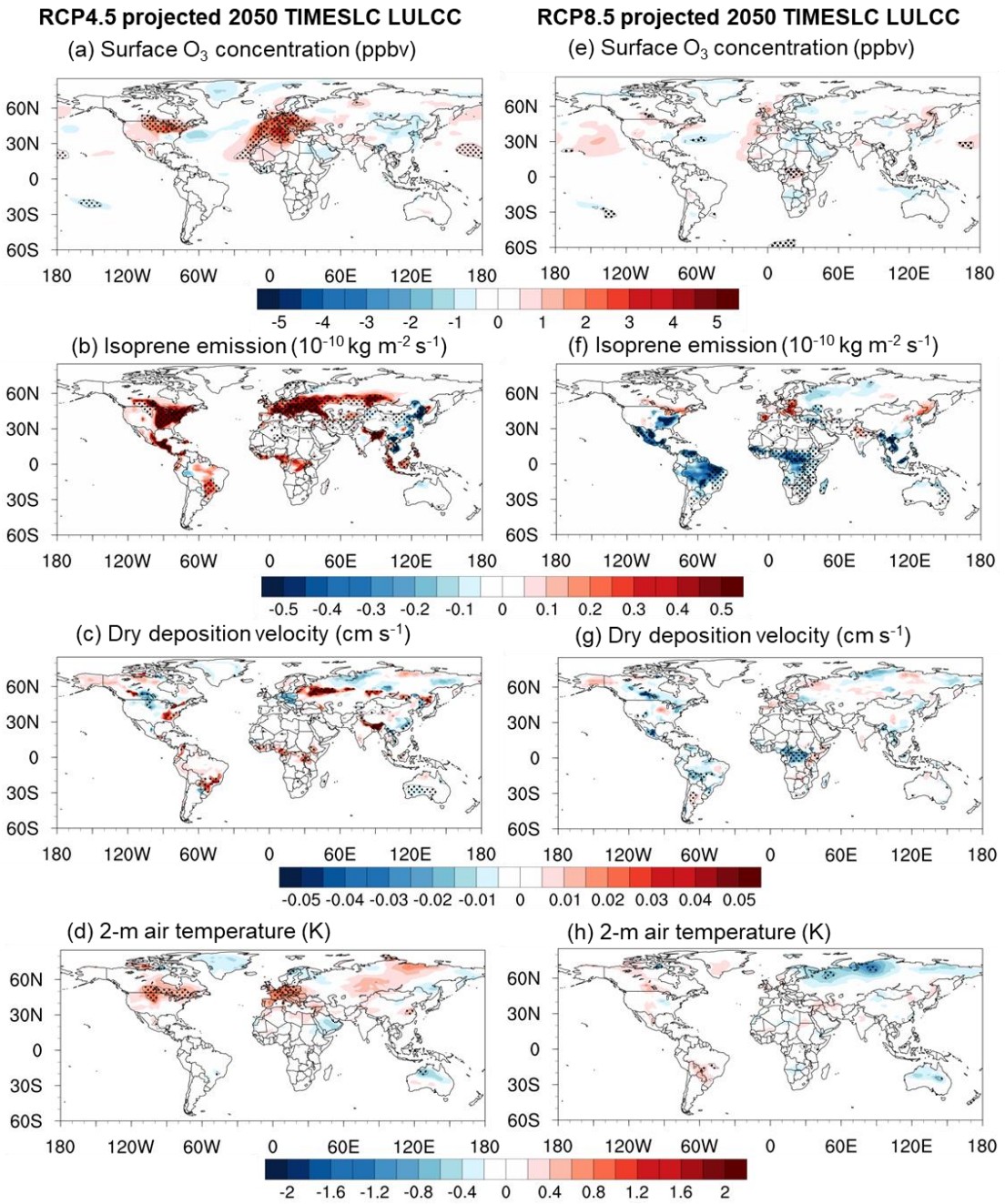



Figure 4. Simulated 2000-to-2050 changes in surface ozone, isoprene emission, dry deposition

velocity and 2-m air temperature for the boreal summer averaged over the 30-year analysis window,
under two future scenarios (RCP4.5 and RCP8.5) of LULCC. Regions with dots indicate changes that
are significant at the 95% confidence level. These results are from the On-line runs (land forcing 2050
minus 2000) with dynamic meteorological responses to LULCC from time-sliced simulations On-
line_45TS and On-line_85TS (Table 1).

Changes in isoprene emission also correlate with temperature changes (Figs.

4b, d; Figs. 4f, h, Fig. S2b). Isoprene emission also increases in regions with forest
expansion, reflecting not only the biogeochemical effects due to higher fractional
coverage of isoprene-emitting vegetation types (Section 3.2), but also the
biogeophysical effects arising from changing 2-m air temperature.

Changes in dry deposition velocity (Figs. 4c, g, Fig. S2c) also correlate to

meteorological changes. In the dry deposition scheme, stomatal resistance can
respond to atmospheric dryness and soil water stress. For instance, drier conditions
are captured in RCP4.5 in the north-central US as initiated by the LULCC further
east, with anomalous moisture divergence (Fig. 5n) and soil moisture (Fig. 5o). The
drier conditions could result in suppressed dry deposition in the corresponding regions
(Fig. 5c). The responses of dry deposition to drought conditions have also been
observed by recent studies (e.g., Lin et al., 2019). Furthermore, changes in surface
roughness can influence aerodynamic resistance and thus dry deposition via
modifying boundary-layer turbulence. In LULCC scenarios, surface roughness is
modified substantially with increases in RCP4.5 (Fig. 2j) and reductions in RCP8.5
(Fig. 2o), which generally decrease (increase) resistance and enhance (decrease) dry
deposition in RCP4.5 (RCP8.5) in LULCC regions, though the overall changes in dry
deposition is dominantly shaped by the combined meteorological effects of LULCC.

Table 2 shows in general, the percentage changes in isoprene emission and dry

deposition in the On-line simulations are smaller than in the Off-line simulations in
both scenarios, reflecting that on a global scale, LULCC-induced meteorological
changes partly offset the biogeochemical effects of changing land cover types on
ozone.
Thus, changes in ozone can be caused by both biogeochemical and
biogeophysical effects of LULCC; furthermore, both effects are highly coupled with
each other. We find that in particular the biogeophysical effects of LULCC play
critical roles in modulating surface ozone. Hereafter, we focus on the broad regions of
North America and Europe, in order to elucidate the origins of surface ozone changes
in response to LULCC-induced meteorological changes. We also focus on RCP4.5
only, because no significant changes in ozone or other meteorological variables are
found for the RCP8.5 LULCC scenario.

3.3.1 North America under RCP4.5 reforestation
For RCP4.5, North America is subjected to intensive regional changes in the
land cover over the eastern US and southern Canada (Fig. 5d). Significant changes in
surface ozone (Fig. 4a) and 2-m air temperature (Fig. 4d) are found over large
continuous areas in North America, including both the regions with intensive LULCC
and regions where LULCC is minimal. Let us first focus on the forested regions with
intensive LULCC (Fig. 5d), where reforestation results in a significant decrease in
surface albedo (Fig. 5e). In the boreal and temperate mixed forests of southern
Canada and northeastern US, such an albedo reduction results in a substantial
enhancement in absorbed solar radiation (Fig. 5g). Typical of these forest types, the
enhanced net radiation is in turn largely dissipated by higher sensible heat (Fig. 5h)
instead of latent heat (Fig. 5i), resulting in a 0.5–1 K rise in average air temperature
(Fig. 5j). This generates a warmer and drier boundary layer with suppressed
precipitation (Fig. 5k), cloud cover (Fig. 5l), and soil moisture (Fig. 5o), constituting a
feedback that likely further enhances net radiation. All these meteorological changes
contribute to higher surface ozone concentrations (Fig. 5a) beyond the
biogeochemical effects alone. In southern Canada, the drier conditions even help
suppress dry deposition (Fig. 5c), further enhancing ozone there. These
biogeophysical effects can be summarized by the cross-amplifying pathways in the
blue box in Fig. 1. Furthermore, reduced wind speed (Fig. 5m) following enhanced
roughness (as represented by vegetation height in Fig. 5f) may also reduce moisture
transport to these forests, inducing a greater moisture divergence there (Fig. 5n).

In contrast, in the subtropical broadleaf forests in the southeastern US,

enhanced forest cover and albedo instead lead to greater moisture convergence from
the Gulf of Mexico (Fig. 5n). This generates more favorable water conditions that not
only dampen meteorological changes there but also promote dry deposition, leading
to only slight changes in ozone. These can also be seen in the cross-counteracting
pathways in the blue box of Fig. 1.

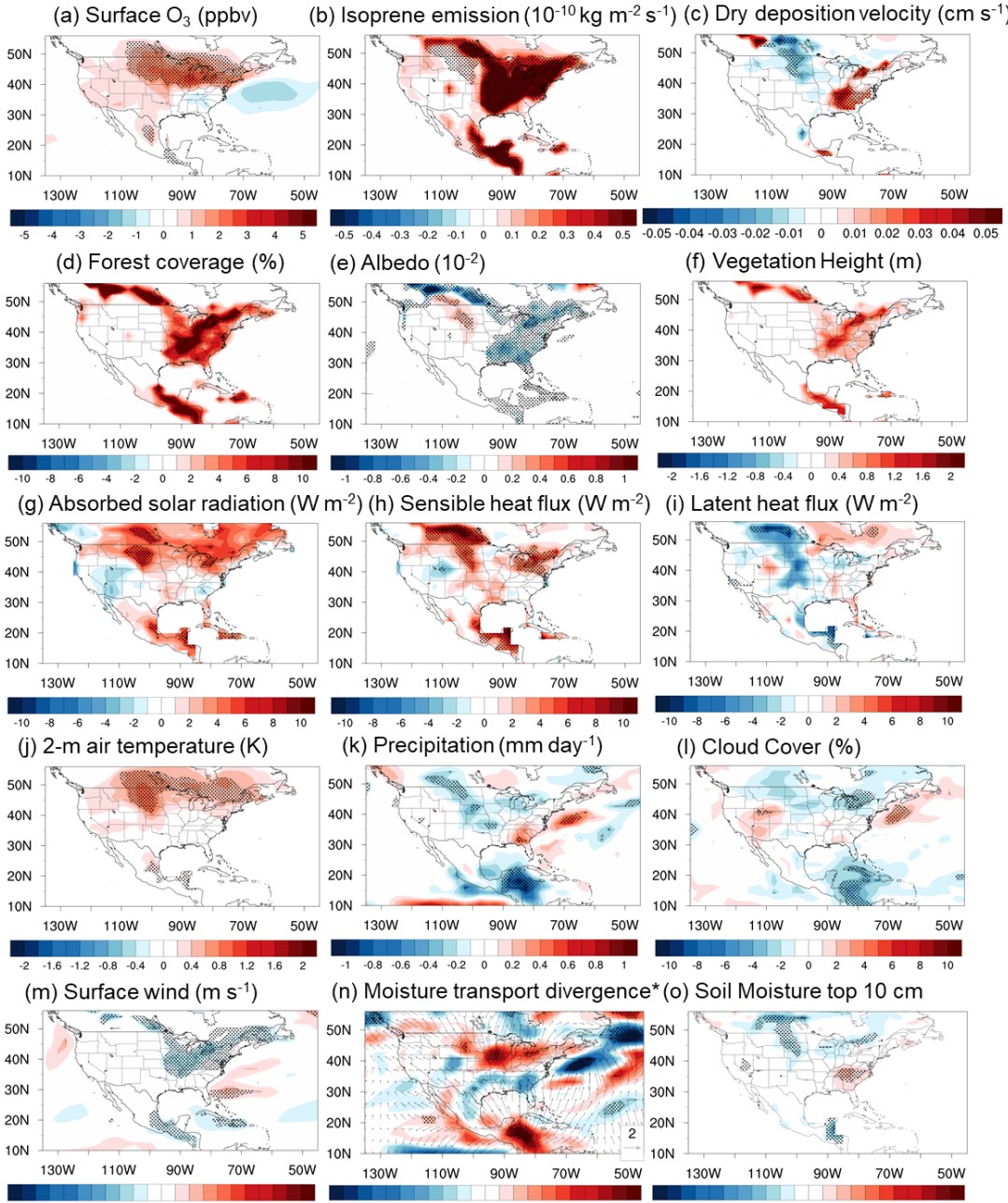

Figure 5. Changes in surface ozone, isoprene emission, dry deposition velocity, projected forest, simulated surface albedo, vegetation height, surface net solar radiation, sensible and latent heat fluxes, 2-m air temperature, precipitation, cloud cover, surface wind, vertically integrated moisture transport divergence (vector: $kg\ m^{-1}\ s^{-1}$, shading: $10^{-5}\ kg\ m^{-2}\ s^{-1}$), and soil moisture at top 10-cm layer during the boreal summer over North America due to 2000-to-2050 RCP4.5 projected LULCC. Regions with dots indicate changes that are significant at the 95% confidence level.

Surface ozone also increases significantly over the locations where land use
does not change significantly, especially over the Midwest and Great Plains regions of
north-central US (Figs. 5a and 5d). The ozone enhancement is found to correspond to
the drier, warmer and sunnier conditions there that can be considered as "remote
effects" of LULCC. Such conditions are associated with enhanced moisture
divergence (Fig. 5n), which could be caused by the stronger convergence over the
surrounding reforested regions that diverges moisture flow from the Great Plains, as
well as reduced surface wind speed (Fig. 5m) that can influence regional moisture
transport to these regions. The vertically integrated moisture fluxes at present-day
conditions are shown in Fig. S3a, illustrating that normally moisture transport from
the Gulf of Mexico is deflected by the Rocky Mountains and toward the eastern and
north-central US. Due to reforestation, moisture transport is deflected further east and
it generates an anomalous moisture flux divergence around the Midwest and Great
Plains, resulting in drier conditions in these regions. The drier and warmer boundary
layer are also reflected by the lower precipitation (Fig. 5k), cloud cover (Fig. 5l), soil
moisture (Fig. 5o), latent heat (Fig. 5i), and the associated higher net radiation (Fig.
5g), sensible heat (Fig. 5h) and air temperature (Fig. 5j). The lower soil moisture can
also reduce dry deposition there (Fig. 5c). All these changes can act together to
enhance surface ozone over the north-central US as remote effects of LULCC
elsewhere; these pathways can be summarized by the yellow box in Fig. 1.

3.3.2 Europe under RCP4.5 reforestation
Substantial increases in surface ozone (Fig. 6a) and air temperature (Fig. 6j)
are found in Europe due to the RCP4.5 LULCC scenario, whereby substantial
reforestation occurs over in the boreal and temperate mixed forests in the European
continental regions (Fig. 6d), modifying surface energy balance significantly. Over
the regions with intensive LULCC, the biogeophysical pathways shaping boundary-
layer meteorology and ozone are largely similar to southern Canada and northeastern
US, where the forest types are similar (see blue box in Fig. 1). In brief, reduced
albedo (Fig. 6e) leads to enhanced net radiation (Fig. 6g) and sensible heat (Fig. 6h),
raising 2-m air temperature over a large area by 0.4–1.2 K (Fig. 6j), and constituting a
hydrometeorological feedback that reduces precipitation (Fig. 6k), cloud cover (Fig.
6l), and soil moisture (Fig. 6o). These changes generate warmer, drier and sunnier
conditions over the forests that favor higher ozone levels. Reforestation also decreases
surface wind speed (Fig. 6m) and moisture transport at the near-surface level.

The increases in surface ozone are also found to extend westward and

southward beyond the regions with intensive LULCC, likely reflecting remote effects
(Fig. 6a). The lower-level wind patterns at 850 hPa under present-day conditions are
shown in Fig. S3b, showing that reforested regions are originally on the southerly
branch (eastern part) of the Azores High anticyclone. Circulation changes in response
to reforestation appears to enable the anticyclonic system to extend eastward,
allowing sunny and warm conditions typical of the Azores High to prevail over much
of western Europe and parts of North Africa, and enhancing surface ozone there.

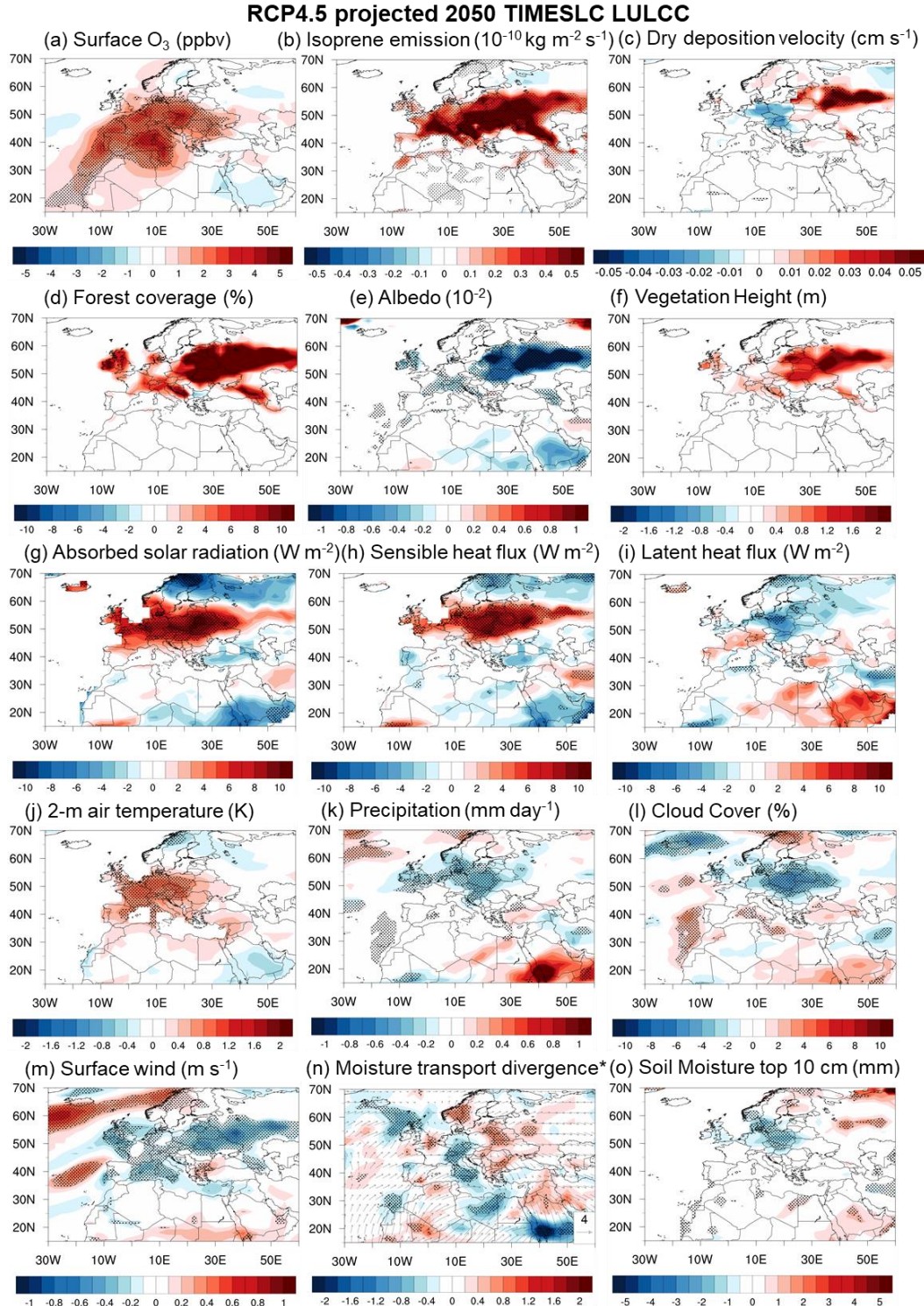

**RCP4.5 projected 2050 TIMESLC LULCC**

(a) Surface $O_3$ (ppbv)    (b) Isoprene emission ($10^{-10}$ kg m$^{-2}$ s$^{-1}$)    (c) Dry deposition velocity (cm s$^{-1}$)

(d) Forest coverage (%)    (e) Albedo ($10^{-2}$)    (f) Vegetation Height (m)

(g) Absorbed solar radiation (W m$^{-2}$)    (h) Sensible heat flux (W m$^{-2}$)    (i) Latent heat flux (W m$^{-2}$)

(j) 2-m air temperature (K)    (k) Precipitation (mm day$^{-1}$)    (l) Cloud Cover (%)

(m) Surface wind (m s$^{-1}$)    (n) Moisture transport divergence*    (o) Soil Moisture top 10 cm (mm)

Figure 6. Similar to Fig. 5 but for Europe under RCP4.5.

Overall, we find that biogeophysical effects can have strong impacts on surface ozone through modifying local and remote meteorological conditions such as surface warming, drying and circulation anomalies initiated by local LULCC (Fig. 1).

Our results of temperature changes are consistent with the previous study of Swann et
al. (2012) that illustrated the local and remote climate effects of the northern
midlatitude reforestation. They conducted a model experiment with extreme
afforestation, and found substantial warming in North America and Europe. In
addition, Govindasamy and Caldeira (2001) and Unger (2014) also found surface
cooling due to deforestation.

3.3.3 Transient experiments versus time-slice experiments

In the above sections, for a direct, parallel comparison with the Off-line

configurations, we have used the time-sliced experiments with the present-day land
cover in year 2000 and future land cover in year 2050. However, in reality the
LULCC is transient with the land cover changing gradually; therefore, transient runs
in On-line mode with the land cover evolving from the present-day all the way to year
2065 are also conducted (On-line_45 and On-line_85, each with two ensemble
members; see Table 1). Fig. 7 shows the changes in ozone and other variables from
the transient simulations, using 2036 to 2065 as the 30-year averaging period to
capture interannual variability. We find that changes in ozone, 2-m air temperature,
and other factors controlling ozone are very similar between the transient and time-
sliced runs (see also Table 2), with only statistically insignificant differences in
different variables in most places (see Fig. S6 in the supplement). The consistent
simulated results from the transient (Fig. 7) and time-sliced (Fig. 4) LULCC further
reflect the robustness of the LULCC-induced signals at least over North America and
Europe, which are strong enough to cause changes in meteorology and ozone
pollution in places remote from LULCC, and indicate that the atmospheric responses
and biogeophysical effects are generally fast-responding at a quasi-steady state on
timescales of years to decades with respect to the slow LULCC.

Figure 7. Similar to Fig. 4 but these results are from the transient simulations On-line_45 and On-
line85 (Table 1), averaged over the two ensemble members for each scenario.

**4. Conclusions and Discussion**


LULCC is expected to continue to co-occur with future socioeconomic

development and anthropogenic emission reduction strategies. These changes likely
had, and will continue to have a large impact on air quality and climate. However, the
impacts of LULCC on surface ozone pollution are not fully understood, and the
attribution to different LULCC-mediated pathways is far from complete. Here, we
investigate and quantify specifically the biogeochemical effects (via modifying
ozone-relevant chemical fluxes), biogeophyscial effects (via modifying the overlying
meteorological environment), and the combined effects of LULCC on surface ozone
air quality.

We address the biogeochemical effects alone by performing CESM

simulations with prescribed meteorology, and investigate the combined effects using
atmosphere-chemistry-land coupled configuration with dynamic meteorology. We
find that the biogeochemical effects of changing isoprene emission and dry deposition
following LULCC mostly offset each other, resulting in only modest changes in
ozone by up to 2 ppbv from 2000 to 2050. However, surface ozone can be
significantly altered by up to 5 ppbv when considering the combined effects
associated with the LULCC. In particular, the biogeophysical effects facilitated
through temperature changes plays a critical role in shaping surface ozone. We find
that surface ozone changes correspond well with temperature changes in RCP4.5 over
both regions with intensive LULCC and regions with limited LULCC.

The surface ozone changes due to future LULCC are comparable with

anthropogenic emissions and climate, and thus should be taken into account in future
research and policy planning. For example, summertime surface ozone changes
induced by climate change alone are projected to increase by 1–10 ppb in the US,
Europe, East and South Asia (e.g., Jacob and Winner, 2009; Fiore et al., 2012). It is
also found that the combined effects of changing climate, emissions and land cover on
surface ozone are up to 10 ppb in the US under two RCP scenarios, and the
contributions from the three factors have comparable magnitudes although of
different signs (Val Martin, et al., 2015). Wang et al. (2011) found that in China,
summertime surface ozone decreases by ~10 ppb on average with a maximum
reduction of 25 ppb if all anthropogenic emissions are removed. Our simulated ozone
changes induced by LULCC are substantial and within the same order of magnitude
as the above studies and others that considered meteorological responses to LULCC
(Ganzeveld et al., 2010; Val Martin et al., 2015). This highlights the important roles
of LULCC in modulating surface ozone.

The mechanisms behind hydrometeorological responses to LULCC are

summarized in Fig. 1. In brief, first, surface properties and processes (e.g., surface
albedo and evapotranspiration) are altered, leading to changes in the surface energy
balance. In boreal and temperate mixed forests, the albedo effect dominates, leading
to higher net radiation, sensible heat and surface temperature, but reduced
precipitation, cloud cover and soil moisture. These local changes can also induce a
regional circulation response, in particular the formation of anomalous moisture
divergence and corresponding warmer and drier conditions over the surrounding
regions even with limited LULCC. In subtropical broadleaf forests, however, both the
albedo and evapotranspiration effects are important and they tend to offset each other,
leading to minimal hydrometeorological changes.

In our analysis of LULCC-induced hydrometeorological changes, we have

focused on the surface and the overlying boundary layer. Many studies have found
that LULCC-induced surface changes can propagate to upper levels as high as 200
hPa (e.g., Chase et al., 2000; Swann et al., 2012; Medvigy, et al., 2013; Xu et al.,
2015; Jia et al., 2019). In our study, significant meteorological changes can be
detected at the upper levels up to 200 hPa due to LULCC (not shown), which can lead
to circulation changes, storm track displacement, and anomalous subsidence
especially at midlatitudes, likely constituting feedbacks on precipitation, moisture
transport, and temperature. However, we find no clear conclusions as to whether these
upper-level changes and feedbacks could have sufficient influence on ozone-relevant
hydrometeorological conditions beyond that can be explained by boundary-layer
dynamics alone.
Weaker responses of temperature as well as of surface ozone to LULCC are
found in RCP8.5 compared with those in RCP4.5. The different extent of temperature
responses can be attributed to the location where LULCC occurs. For RCP4.5,
LULCC is most intense in the midlatitude regions of the Northern Hemisphere. In
contrast, most LULCC for RCP8.5 occurs over the equatorial regions and Southern
Hemisphere. Temperature responses to LULCC may be less sensitive to tropical
changes or changes over the Southern Hemisphere that is dominated by the vast
oceanic expanse. Van der Molen et al (2011) using other models also found similar
patterns, and named such climate responses to LULCC as "tropical damping". The
classical theory of such "tropical damping" is associated with a decrease in cloud
cover after deforestation, which then results in increased incoming radiation at the
surface and a lower planetary albedo, both counteracting the increase in surface
albedo with deforestation.
Our study has several limitations. First, the energy transport between the
ocean and land has not been taken into account. Although using a fully interactive
ocean component would increase the variability of simulated climate and decrease the
signal-to-noise ratio in sensitivity experiments using small forcings, such as LULCC
(e.g., Davin and de Noblet-Ducoudre 2010, Brovkin et al., 2013), coupled
atmosphere-ocean simulations are crucial for future climate change projections for the
longer term (e.g., well past the end of the 21st century). In addition, future LULCC
projections in RCPs are predicted from the ensemble of socioeconomic and emission
scenarios to match identified pathways of greenhouse gas concentrations. Large
uncertainties remain in such projections, calling for more skillful design of LULCC-
related metrics and the corresponding spatial patterns for better air quality predictions.
Third, the biogeochemical effects of LULCC on ozone in this study do not consider
climatic changes or anthropogenic emission change, but only focus on the more
immediate effects generated from LULCC such as isoprene emission and dry
deposition, mostly due to model limitations. For example, $NO_x$ emission is projected
to decline sharply over the northeastern US in RCP4.5. As $NO_x$ level decreases, ozone
production may become more $NO_x$-limited and thus the sensitivity to isoprene
emission may be reduced, rendering the overall biogeochemical effects of LULCC
smaller. However, since the biogeophysical effects operate in locations remote from
the source regions, they may be less affected by $NO_x$ emission changes in the source
regions. The full biogeochemical effects of LULCC on ozone that include
biogeochemical cycle-climate feedbacks and co-effects of anthropogenic emission
and LULCC will warrant further investigation but will foreseeably present greater
challenges for process attribution and interpretation.

Atmospheric internal variability is one factor that could affect the significance

of our results. Large internal variability of the climate system reduces the signal-to-
noise ratio for LULCC-induced climatic changes (Deser et al., 2012). To ascertain the
impacts of such variability, we have adopted an analysis period of 30 years for both
the time-sliced simulations (looping over the single-year LULCC forcing) and 2-
member ensemble transient LULCC simulations. Results from both simulation
approaches all show broadly consistent signals induced by LULCC in North America
and Europe, indicating the significance of our results and the strong signal-to-noise
ratios at least over those continents. When applicable, more ensemble members for
transient simulations can be used to further confirm the impacts of such variability.
Furthermore, we have compared the magnitudes of interannual standard deviations of
near-surface temperature of the CTL run with the LULCC-induced climate signals.
Our results show that the climate signals are not weak and can be regionally
comparable to interannual variability at midlatitudes (Fig. S4), e.g., over North
America and Europe. It is also noteworthy that the time-sliced experiments with
single-year forcing looped for multiple years give results very similar to the transient
simulations, further pointing to the robustness of LULCC impacts.

Our study highlights the complexity of land surface forcing and the

importance of biogeophysical effects of LULCC on surface ozone air quality,
emphasizing the importance of LULCC in shaping atmospheric chemistry that could
be as important as anthropogenic emissions and climate. Our study can provide
important reference for policy makers to consider the substantial roles of LULCC in
tackling air pollution and climate change, to develop a more comprehensive set of
climatically relevant metrics for the management of the terrestrial biosphere, as well
as to explore co-benefits among air pollution, climate change and land use
management strategies.



**Author Contribution**

L. Wang designed the model experiments, performed numerical simulations
and analysis, and co-wrote the manuscript; A. P. K. Tai and C.-Y. Tam are the co-
principal investigators, who designed the research, performed some of the analysis,
and co-wrote the manuscript; and all authors contributed to the interpretation of the
results and writing of the paper.

**Acknowledgments**

This work was supported by the Vice-Chancellor Discretionary Fund (Project
ID: 4930744) from The Chinese University of Hong Kong (CUHK) given to the
Institute of Environment, Energy and Sustainability. It is also supported by a General
Research Fund grant (Project ID: 14306015) from the Research Grants Council of
Hong Kong given to A. P. K. Tai.

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
