# Peer review of "Impacts of future land use and land cover change on mid-21[st]- century surface ozone air quality: Distinguishing between the biogeophysical and biogeochemical effects"

_Atmospheric Chemistry and Physics, 2019_

## Referee Comment (RC1) · Anonymous Referee #2 · 26 Dec 2019

This paper advances a new framework for quantifying the influence of land use and land cover (LULC) on surface ozone. The authors separately consider feedbacks labeled biogeochemical, defined as the responses of sources and sinks of ozone through vegetation, and labeled biogeophysical, defined as physical climate feedbacks in response to changing albedo and the responses of surface energy budgets and hydrologic cycling, and thus atmospheric circulation. Two scenarios for LULC out to mid-century are considered, RCP4.5 and RCP8.5. The authors conclude that the impact from biogeophysical pathways is important and can be larger than that from biogeo-

chemical pathways.

General points:

1. It would be useful to add some discussion regarding the potential for low frequency climate variability to influence the findings given that SSTs and sea ice are prescribed, and only a single ensemble member is used. See for example Deser et al., 2012: Deser, C., Knutti, R., Solomon, S. et al. Communication of the role of natural variability in future North American climate. Nature Clim Change 2, 775–779 (2012) doi:10.1038/nclimate1562

2. The treatment of vegetation, specifically what is prescribed versus calculated in the model should be clarified in the text. For example, LAI is described in Line 192 as being prescribed from observations, yet Figure 2 shows changes under the LULC scenario. Are these future changes prescribed as part of the LULC scenario or is the vegetation changing dynamically?

3. What are the assumptions underlying the number or fraction of isoprene emitters in the forest in the land model? Is it assumed that the emissions of isoprene from a forest are constant over time and globally in the model? Similarly, what assumptions underlie the treatment of dry deposition in the model?

4. Is anything else besides LULC changing albedo in the model? This seems to be the case in Figure 5b (and 8b?), where there is no corresponding change in Figure 2. Is snow cover changing dynamically in the model? Some discussion is needed. It could be useful to show the change in cloud cover too.

5. It first appears in Figure 4 that LULC is larger in RCP4.5 than in RCP8.5, which becomes more apparent later, and is briefly mentioned in the text (Lines 523-524). It would be clearer to include some discussion of this when these results are first displayed. Furthermore, are LULC changes in these two RCP scenarios consistent with the assumptions for greenhouse gases and other emission changes? Or is LULC decoupled from choices about other emissions?

6. The conclusion and discussion section is somewhat redundant with earlier text. I suggest shortening to focus on the most important messages of the paper and their implications.

7. The figures are small and hard to read.

Detailed points:

8, Lines 72-74. What about direct reaction of O3 + HOx and non-stomatal pathways for deposition?

9. Lines 102-104, and Figure 4, and elsewhere: Are these annual mean values or summertime?

10. Line 238. What are the time periods considered for present and future?

11. Line 405. Doesn't NOx decline sharply in the RCP scenarios in this region? Emissions are held constant in the model to isolate LULC changes, but some acknowledgment that the full scenario would have large ozone changes due to precursor emission changes seems warranted. Do the LULC changes amplify or dampen the emission driven changes?

12. Line 408. Does this chemistry require updating in light of newer work indicating sufficient OH recycling at low NOx levels? At minimum, some discussion is needed.

13. Line 441-443. This discussion is qualitative when it should be possible to quantify the findings. For example, why not report a spatial correlation to strengthen this point? Also, are the dry deposition and isoprene emission patterns the same in the offline and online versions? From Table 2, I expected more damping of the responses in the offline version but it's really hard to compare Figures 3 and 4 beyond looking at patterns. Improving the figures to enable the reader to extract more meaningful and detailed information would be helpful.

14. Line 519. Where do we see that soil is drier?

15. Line 544. Why not show the same evidence as for Figure 6, perhaps in the supplement?

16. Line 550. Should this be RCP4.5 here? Otherwise this sentence does not make sense.

17. Line 555-556. This discussion is unnecessarily speculative as it should be possible to demonstrate whether this mechanism is operating in the model or not.

18. Line 560-562. Would be helpful to refer to Figure 2 here.

19. Line 572. How have local responses been separated out here? Wasn't LULC changed everywhere in the model at the same time?

20. Line 586. Could refer back to Table 2 here.

21. Line 622-623. Briefly explain how this conclusion was reached. Are these the only mechanisms by which the model can respond to changes in LULC?

22. Figure 1. Some additional detail could be added here. For example, in the green box, won't the sign of the change depend on the location dependent to how the westerlies are displaced? Similarly, in the yellow box, should the sign of the change depend on local chemistry and emissions in the case of isoprene, and also on location with respect to the anomalous high and changes in moisture divergence or net surface radiation?

23. Table 2. Why not add columns for the changes in surface ozone here?

24. Figure 4. How has significance been assessed? This should be explained in the methods section.

---

## Referee Comment (RC2) · Anonymous Referee #3 · 17 Feb 2020

Wang et al. aimed to address the impact of future land use and land cover change (LULCC) on surface ozone. They authors differentiated between the role of biogeo-chemical effects and biogeophysical effects by conducting fixed-dynamics simulations and coupled chemistry-climate simulations using a 3-D global model. They found that the biogeochemical effects are relatively small due to the counteracting impacts of iso-prene emission and ozone uptake by plants. The biogeophysical processes likely play a more important role both at local scale through albedo effects and surface energy redistribution, and at regional scale through teleconnections. The manuscript is well

structured and clearly written, and fits well into the scope of ACP. Below are some minor issues to be considered before publication.

Throughout the manuscript, I do not see a discussion on whether the changes in ozone are significant, compared to the perturbations caused by natural climate variability. E.g. Fig 3 needs to include a confidence level. I see some dots in other figures but there are no explanations. The authors may also add some explicit discussion on ozone changes induced by anthropogenic emissions under RCP4.5 and RCP8.5, to give the readers a clearer idea what the LULCC impacts are compared to emissions and climate.

L130, L131 and many other places: LULC should rather be land use and land cover change (LULCC). Land cover itself cannot 'induce' anything. I'd suggest to check through the manuscript and use LULCC instead of LULC.

L221: CAM4 should be "On-line mode of CAM4-Chem-CLM".

L225: I think you prescribed anthropogenic emissions, biomass burning and long-lived species CO2, CH4 and N2O for ALL simulations, not just the 5 coupled simulations, right? Combine L224-225 with L212-213 together and provide more details, e.g. what inventories/values you used.

Fig. 3g, k, h, l: these plots are not clear. Perhaps change the scale of color bars.

L408-409: Shouldn't the tropical region (e.g. the Amazonia) be the same regime as the southeast US? With increases in isoprene emission that consumes more ozone and increases in dry deposition, I assume ozone will decrease as in the southeast US. This is not shown in Fig 3g, why? Again, change the color bar may improve the visualization, and adding a map showing relative changes in percentage in the supplement would help.

L432: Give a rough number/range of ozone responses to changes in anthropogenic emissions and climate here.

L452-458: This paragraph is not clear. It reads like the changes in isoprene emission and dry deposition are due to meteorological changes (warmer temperature, drier/wetter conditions), but actually you cannot differentiate whether the changes are directly caused by LULCC or LULCC-induced meteorological changes, right? You mentioned that isoprene emission changes are smaller than the off-line values, suggesting meteorology partly offsets the direct LULCC impacts. But intuitively, reforestation leads to warming (mid-latitudes in Fig 4d) and then more isoprene emission, so the LULCC-induced meteorology changes add on to the direct LULCC impact on isoprene?

L462-463: Avoid using the terms "biogeophysical" and "biogeochemical" here. You are referring to impacts on isoprene emission, not ozone.

L508-514: How will the anomalous high with low-level divergent wind in RCP4.5 experiment influence ozone advection and transport?

L538: What's the meaning of "a positive tendency" for the energy budget?

L576-578: Also add Unger 2014 Nat. Clim. here for the cooling of deforestation.

—————————————————

---

## Referee Comment (RC3) · Anonymous Referee #4 · 20 Feb 2020

This paper looks at the biogeochemical and biogeophysical impacts of land-cover change on surface ozone. Surface ozone perturbations are much larger when including the biogeophysical changes, than when looking at the biogeochemical aspects alone. This is an interesting result and could be published.

However, the paper suffers due to methodological and conceptual errors. Considering the internal variability of the atmosphere, the authors need to do considerably more work to show that the biogeophysical ozone signal is due to changes in the land cover instead of internal variability. This paper might be publishable after extensive revisions.

[Figure]

1. I believe the authors may have looked at the statistical significance of their difference maps (the figures appear to show some cross-hatching), but they are difficult to read and are not mentioned in the text. Statistically significant areas need to be highlighted and the significance level discussed in the text. In many cases non-significant signals are discussed. They should not be. For example, there is extensive discussion of the ozone signal in Europe, but judging from Figure 4 these changes are not significant.

2. The response of the atmosphere to the land-surface is complex. It is not as simple as simply applying the thermal wind balance to surface temperature perturbations. If the authors do wish to include the cause and effect of the atmospheric perturbations to land-cover change they would do well to enlist the help of an atmospheric dynamicist. As it stands he paper will only be strengthened by omitting the rather simplistic meteorological explanations of the impact of land-cover change on the atmosphere.

There have been many simulations of the atmosphere to perturbations of surface temperatures (noting the response is much different in the tropics than the mid-latitudes). The authors could support their hypothesis by citing relevant papers. On the otherhand, many studies do show a response in the general circulation to changes in landcover (e.g., see Lague et al. [2019, preprint DOI, 10.31223/osf.io/dbyqu] and references therein.) I am struck by the very similar northwards displacement of the jetstream in the RCP4.5 and RCP8.5 simulations, despite different landcover changes and changes in the surface temperature response. This may argue for similar changes in the overall circulation. These changes appear to be on the hemispheric scale. It is unclear if any of the local changes are really significant.

3. In their interpretation of the response to landcover change the authors should be mindful of the large internal variability of the atmosphere. Even where the differences are found to be statistically significant the interpretation of these differences to changes in land-cover (instead of internal variability) may be problematical. The differences in the simulations could be simply due to decadal variability in the atmosphere. As shown in Deser et al. (2012) [Clim Dyn (2012) 38:527–546DOI 10.1007/s00382-010-0977-x],

for example, in most places it takes more than 30 ensembles of 10 year average differences in transient simulations (e.g., 2028–2037minus 2005–2014) to see significant differences in precipitation. While the present simulations might have less variability due to the fixed sea-surface temperatures it is unclear to me how much this reduces the variability. The presence of internal variability may obfuscate any signal from the change in land surface. For example, if I take the stipples in Figure 6 as gridpoints with significant differences (?) there are many regions of stipples throughout the world (even in parts of the S.H.) which seem significant. If one takes a significance level of 95%, this suggests 5% of the points may only appear to be statistical different.

However, the timeslice experiments in essence add another ensemble member. Similarities between the timeslice analysis and the transient analysis may point to robust differences. The authors need to do more work to attribute the changes to changes in land-cover. Their meteorological attribution, as described above, is probably not correct.

Minor Comments:

1. I felt the paper could be better referenced. Please back up with more references, e.g. L106-108 "Dry deposition. . .." L113-114 "The dry deposition. . .. L236-237

and other locations. . ..

2. It is probably important to emphasize somewhere that the impact of the surface on the atmosphere is complex. For example, taking Lague et al. (2019, preprint DOI, 10.31223/osf.io/dbyqu) as an example, the impact can be through changes in albedo, evaporative resistance, and surface roughness. How these play out together and interact with each other is not simple. These changes may impact the clouds, boundary layer turbulence etc. A short paragraph explaining these influences might be in order in the introduction. I don't think the paper mentions surface roughness anywhere. In addition, the explicit impact of the surface on the boundary layer should be discussed as this will impact the dry deposition of ozone and the mixing and venting of ozone in

the boundary layer.

3. Figure 1. While the land surface may influence the upper troposphere, the exact connection is not really clear. In addition, the tropical response (where there is no real jet-stream) is likely much different than the mid-latitude response.

4. Data and Methods Section. Prior to going into the model specifics, it might be useful to give a broad overview of the simulations. For example, it was confusing when the paper first discussed online and offline simulations and the setup of both. Note also the online and offline simulations likely will have very different boundary layers with different clouds and radiation so a comparison of these two model setups is not straightforward (e.g., Brownsteiner et al., 2015).

5. Could the authors clarify the difference in dry-deposition in the off-line land cover change simulations? Are the differences shown only due to differences in stomatal conductance. Is the dry deposition also sensitive to LAI or the type of vegetation even without considering stomatal conductance? I assume the parameterized boundary layer turbulence is the same in each case, correct?

6. L641 "vice versa", please spell out.

7. L357, Do the isoprene emissions depend on the makeup of the forest expansion?

8. It is unclear why NOX is shown in Figure 3. I don't think it is discussed.

9. In discussing the biogeophysical response the authors did not discuss changes in the ozone deposition velocity. This seems to be a field that would be easy to show and could be more directly attributable to land cover change.

10. Figure 4 and other figures showing the difference between online model simulations. I believe these figures show the difference between the final 10 years of the transient simulation and 10 years from the on-line CTL. Please state explicitly (maybe in the figure caption?).

11. In discussing figure 4, the authors mention the correlations between difference fields. In each case please give the correlation coefficient between the fields and its significance. I would suspect that in many cases these correlations are not actually significant, in which case the authors need to refine their language in discussing the relation between the fields.

12. L457 "meteorological changes", these would include not only stomatal response which is part of the story, but the impacts of surface roughness and surface heat exchange on boundary layer turbulence.

13. L502 -L520 (also L542-L550). Please delete. This is rather speculative. The response to land forcing is likely to be complex. The argument concerning the thermal wind relation and the jet-stream is pretty "hand-wavey" and I doubt it is correct. Has anyone else seen this? The changes are most likely dynamically consistent with each other (as described in the paragraph), but this is much different than arguing that they are due to changes in the land surface and in particular through the mechanism described.

14. L522 What is the correlation coefficient?

15. Fig. 5 and Fig. 7 and Fig 8. The discussion would be clearer if you put in a panel showing surface ozone (I don't think you need to show the topography in Fig. 5; in Fig. 8 most of the changes don't appear significant).

16. L555-556, "suggesting"... This is doubtful. India is in the subtropics. The atmospheric response to land cover change is likely to be rather different than in the midlatitudes.
* * *

---

## Author Comment (AC1) · 28 Apr 2020

Author Responses to Referees' Comments on **"Impacts of future land use and land cover change on mid-21st-century surface ozone air quality: Distinguishing between the biogeophysical and biogeochemical effects"** by L. Wang et al. (MS No.: acp-2019-824)

Our point-by-point responses are provided below. The referees' comments are *italicized*, our new/modified text is highlighted in **bold**. The revised manuscript with tracked changes is also included in the linked file below for the Editor's easy reference:

https://www.dropbox.com/scl/fi/9prdbqnfqgzup7r65ybu4/Impacts-of-future-land-use-and-land-cover-change-on-mid-21st-century-surface-ozone-air-quality_Prepared_for_ACP_FinalResponse_trackchanges.docx?dl=0&rlkey=raypjbh40m50a4q11q2d1lwek

Responses to Referee #1

*It would be useful to add some discussion regarding the potential for low frequency climate variability to influence the findings given that SSTs and sea ice are prescribed, and only a single ensemble member is used. See for example Deser et al., 2012: Deser, C., Knutti, R., Solomon, S. et al. Communication of the role of natural variability in future North American climate. Nature Clim Change 2, 775–779 (2012) doi:10.1038/nclimate1562.*

> We thank the reviewer for the very helpful comments. The paper has been revised substantially to address the reviewer's concerns point by point, and all changes are cited and discussed in the responses below.

> We added one paragraph discussing the internal variability:

> P39-40, L747-769: "**Atmospheric internal variability is one factor that could affect the significance of our results. Large internal variability of the climate system reduces the signal-to-noise ratio for LULCC-induced climatic changes. This is analogous to the problem of long-term low-frequency variability of the extratropical circulation affecting the interpretation and extraction of climate change signals, especially if short time series (e.g., ~10 years) are used (Deser et al., 2012). When applicable, multiple-member ensemble runs are required to ascertain the impacts of such variability. Our model experiments with a 55-year transient integration with prescribed sea surface temperature and sea ice are not designed to address such low-frequency climate variability. We note however that our climate simulations focus on land-atmosphere biogeophysical interactions, which typically operate on a shorter timescale, and thus the LULCC-induced climate signals that we detected are expected be present when superimposed upon any long-term trajectory and low-frequency variability undergone by the climate system. For land-atmosphere interactions, high-frequency interannual variability on a decadal timescale may be more relevant. To assess the potential impacts of the internal variability of the system on a decadal timescale, we have compared the magnitudes of interannual standard deviations of near-surface temperature of the CTL run with the LULCC-induced climate signals. Our results show that the climate signals are not weak and can be comparable to interannual variability at midlatitudes (Fig. S4), e.g., over North America land areas at ~45°N and also north of 60°N. It is also noteworthy that the time-sliced experiments with single-year forcing looped for multiple years, give results very similar to the transient simulations, pointing to the robustness of LULCC impacts.**"

> .

CTL Standard Deviation over the last 10 yrs

(a) Temperature (degree)

[Figure]

RCP4.5 projected 2050 LULC changes

(b) Temperature (degree)

[Figure]

RCP8.5 projected 2050 LULC changes

(c) Temperature (degree)

[Figure]

Figure S4. Standard deviations of near-surface air temperature in CTL run (a), compared with changes induced by LULCC from On-line runs under two future scenarios: RCP4.5 (b) and RCP8.5 (c) during the boreal summer.

*2. The treatment of vegetation, specifically what is prescribed versus calculated in the model should be clarified in the text. For example, LAI is described in Line 192 as being prescribed from observations, yet Figure 2 shows changes under the LULC scenario. Are these future changes prescribed as part of the LULC scenario or is the vegetation changing dynamically?*

We now add these points to clarify that the LAI in both future scenarios and the present-day case is prescribed:

P10, L206, "We use the Satellite Phenology (SP) mode of CLM4.5 **for all simulations**,".

P18, Figure 2's caption, we added "**The treatment of vegetation including PFT fractional coverage, LAI and canopy height is prescribed using the SP mode of CLM4.5 in both the present-day case and future LULCC scenarios. For the future cases, PFT fractional coverage is derived according to the RCP land use scenarios.**".

P15, Table 1, the last column "**-All simulations use the SP mode in CLM.**"

*3. What are the assumptions underlying the number or fraction of isoprene emitters in the forest in the land model? Is it assumed that the emissions of isoprene from a forest are constant over time and globally in the model? Similarly, what assumptions underlie the treatment of dry deposition in the model?*

The scheme to calculate isoprene emission is the Model of Emissions of Gases and Aerosols from Nature (MEGAN) version 2.1 (Guenther et al., 2012) in the land model. The scheme includes the major known processes controlling isoprene emission from terrestrial ecosystems, such as effects of temperature, solar radiation, soil moisture, leaf age, $CO_2$ concentrations, and vegetation species and density. Thus, isoprene emission is allowed to respond to changes of PFT over time and regions, and to any meteorological changes induced.

The dry deposition velocity is computed based on the multiple resistance approach of Wesely (1989), updated by Emmons et al. (2010), Lamarque et al. (2012) and Val Martin et al. (2014). In the scheme, dry deposition velocity is the inverse of aerodynamic resistance (Ra), sublayer resistance (Rb) and bulk surface resistance (Rc), whereby Rc includes a combination of resistances from vegetation (including stomatal resistance), lower canopy, and ground with specific values for different PFTs. Correspondingly, dry deposition velocity is allowed to respond to primarily meteorological and ecophysiological conditions.

More information can be found in P10-11. We now add some points to clarify:

P10, L216-218: "**Thus, isoprene emission is allowed to respond to spatiotemporal changes in PFTs and the associated changes in meteorological conditions in this study.**"

P11, L224-225: "Correspondingly, dry deposition velocity in the scheme responds to primarily meteorological and ecophysiological conditions; ."

P15, Table 1, the last column:

"**-Isoprene emission is from MEGAN**"

"**-Dry deposition velocity is based on Wesley (1989) updated by Val Martin et al. (2014)**"

*Is anything else besides LULC changing albedo in the model? This seems to be the case in Figure 5b (and 8b?), where there is no corresponding change in Figure 2. Is snow cover changing dynamically in the model? Some discussion is needed. It could be useful to show the change in cloud cover too.*

We checked the snow cover and found that snow cover is rarely changed (shown below). The non-local surface albedo changes are largely caused by the anomalies of precipitation and soil moisture at the top layer, initiated by LULCC. Furthermore, the albedo we investigated here is the surface albedo that changes with surface properties. The cloud cover changes have also been detected (shown below), illustrating similar patterns with the precipitation. For example, cloud cover decreases over the central US and northern India. Such cloud cover decreases, and together with precipitation reduction, drier soil indicates a drier condition in the corresponding areas and a higher surface albedo in the dry condition.

We now add some point to clarify:

P29, L560-561: "Collocated with such a stationary high, there is enhanced (reduced) surface solar radiation (rainfall **and cloud cover**)."

P29, L571-574: "**The drier conditions can also be reflected by the anomalously low precipitation (Fig. 5h), lower soil moisture (Fig. 5i) and the associated higher surface albedo (Fig. 5e), which together with** the anomalous high (Fig. 6e), can all act to promote warming over the central-western US region."

Thank you very much for the valuable suggestions.

[Figure]

Figure: Changes in total cloud cover and snow fraction during the boreal summer over the US (left) and India (right) due to RCP4.5 projected LULCC. Regions with dots indicate that the changes are significant at the 90% confidence level.

*It first appears in Figure 4 that LULC is larger in RCP4.5 than in RCP8.5, which becomes more apparent later, and is briefly mentioned in the text (Lines 523-524). It would be clearer to include some discussion of this when these results are first displayed. Furthermore, are LULC changes in these two RCP scenarios consistent with the assumptions for greenhouse gases and other emission changes? Or is LULC de-coupled from choices about other emissions?*

We now explain it more clearly:

P24, Line 463-465: "**Within the On-line simulations, more substantial responses of temperature as well as of surface ozone to LULCC are found in RCP4.5 compared with those in RCP8.5.**"

LULC projections in these two RCP scenarios are internally consistent with the emission scenarios and development pathways for AR5 of IPCC (Taylor et al., 2012). In general, the RCP4.5 LULC scenario has the most extensive use of land management as a carbon sequestration strategy, with expansion of forest areas and reduction in croplands (Hurtt et al., 2011). The RCP8.5 LULC scenario has the least effective use of land management for climate mitigation, with large expansion in both croplands and grasslands and substantial forest loss. These LULC projections are computed using Integrated Assessment Models (IAM) adopted by the CMIP5 community, incorporating anthropogenic transformation and activities associated with carbon releases (e.g., wood harvest).

The above-mentioned information can be found in the manuscript P12-13, L269-276. We now add brief information to clarify the settings:

P13, L276-279: "**In this study, emissions are held constant at the present-day level for all runs, thus the effects of LULCC can be considered as being decoupled from changes in anthropogenic emissions in order to isolate the effects of LULCC alone.**"

*The conclusion and discussion section is somewhat redundant with earlier text. I suggest shortening to focus on the most important messages of the paper and their implications.*

The discussion has now been shortened.

*The figures are small and hard to read.*

The figures are revised.

*Specific comments:*

*8. Lines 72-74. What about direct reaction of O3 + HOx and non-stomatal pathways for deposition?.*

We have revised the sentence to be clearer:

P4, L72-74: "**The dominant sink of surface ozone is photochemical loss and dry deposition to the surface including to vegetation mainly in the form of leaf stomatal uptake.**"

*9. Lines 102-104, and Figure 4, and elsewhere: Are these annual mean values or summertime?*

The sentence has now been modified to be more accurate:

P5, L97-99: "Heald and Spracklen (2015) estimated the net effect of LULCC under future anthropogenic influences as a decrease of 12–15% in **annual** isoprene emission **globally**."

Figure 4 caption has been revised as: "Figure 4. Simulated 2000-to-2050 changes in surface ozone, isoprene emission, dry deposition velocity and near-surface air temperature with atmosphere-chemistry-land coupled configurations for the boreal summer **averaged over the 10-year analysis window**, under two future scenarios (RCP4.5 and RCP8.5)."

*11. Line 405. Doesn't NOx decline sharply in the RCP scenarios in this region? Emissions are held constant in the model to isolate LULC changes, but some acknowledgement that the full scenario would have large ozone changes due to precursor emission changes seems warranted. Do the LULC changes amplify or dampen the emission driven changes?*

As the reviewer pointed out, anthropogenic emissions of ozone precursors are held constant in the model. In the full scenario with sharp $NO_x$ decline in the northeastern US, ozone chemical regime may shift from the current BVOC-limited regime to the future $NO_x$-limited regime. If the land use and land cover remain the same, the sharp $NO_x$ decline can result in surface ozone decreases. If taking the LULCC and its biogeochemical effects on ozone into account, the reforestation in RCP4.5 can first lead to substantial isoprene increases. If the ozone chemical regime can still remain in the BVOC-limited regime, further increase of isoprene can stimulate surface ozone (as what we found in this study) so as to dampen the emission-driven changes. Once abundant isoprene released from the newly grown trees, ozone chemical regime may shift to $NO_x$-limited regime. Thus, further increase of isoprene can reduce ozone so as to amplify the emission-driven changes. If further taking LULCC biogeophysical effects into account, changes of ozone will be more complex.

Accordingly, we now added some points in the discussion to address these issues:

P38-39, L737-746: "**For example, $NO_x$ emission is projected to decline sharply over the northeastern US in RCP4.5. As $NO_x$ level decreases, ozone production may become more $NO_x$-limited and thus the sensitivity to isoprene emission may be reduced, rendering the overall biogeochemical effects of LULCC smaller. However, since the biogeophysical effects operate in locations remote from the source regions, they may be less affected by $NO_x$ emission changes in the source regions.** The full biogeochemical effects of LULCC on ozone that include biogeochemical cycle-climate feedbacks **and co-effects of anthropogenic emission and LULCC** will warrant further investigation but will foreseeably present greater challenges to process attribution and interpretation."

*12. Line 408. Does this chemistry require updating in light of newer work indicating sufficient OH recycling at low NOx levels? At minimum, some discussion is needed.*

We now add some points:

P23, L433-442: "In contrast, the southeast US is a high-isoprene-emitting region; additional isoprene may react with ozone and $NO_x$, thereby suppressing surface ozone production (Kang et al., 2003; von Kuhlmann et al., 2004; Fiore et al., 2005; Pfister et al., 2008; see also discussion in Section 4). **Furthermore, in the low-$NO_x$ region, OH is largely removed by reactions with biogenic VOCs, producing peroxy radicals that form $HO_2$ or producing organic peroxides. Recent studies found that organic peroxide formation can be reduced; alternatively, these peroxides could be rapidly photolyzed, making them at best a temporary $HO_x$ reservoir (e.g., Thornton et al., 2002; Kubistin et al., 2010). This result implies that in low-$NO_x$ regions ozone production may be $NO_x$-saturated more often than current models suggest.**"

*13. Line 441-443. This discussion is qualitative when it should be possible to quantify the findings. For example, why not report a spatial correlation to strengthen this point? Also, are the dry deposition and isoprene emission patterns the same in the offline and online versions? From Table 2, I expected more damping of the responses in the offline version but it's really hard to compare Figures 3 and 4 beyond looking at patterns. Improving the figures to enable the reader to extract more meaningful and detailed information would be helpful.*

We added a figure of spatial correlations, and also added some points to clarify the patterns between online and offline runs:

P24-25 L469-476, L483-484: "**In particular, ozone changes, together with changes in isoprene emission (Figs. 4b, f) and dry deposition (Figs. 4c) are found in regions without LULCC. Such signals are not captured by the Off-line simulations in which changes only respond to LULCC locally (Fig. 3). On the other hand, changes in surface temperature are found to be correlated well with patterns of changes in ozone (Fig. S2a, d), indicating that the biogeophysical drivers that modify temperature may play critical roles in ozone changes. In the regions where temperature increases, surface ozone increases correspondingly.**"…"**Changes in isoprene emission also correspond closely temperature changes (Figs. 4b, d; Figs. 4f, h, Fig. S2b).**"

We also revised Table 2 to include changes in surface ozone.

Fig. 3 has been revised accordingly to enable the reader to extract more meaningful and detailed information to compare with Fig. 4.

[Figure]

Figure 3. Simulated 2000-to-2050 changes in surface ozone, isoprene emission, and dry deposition velocity under RCP4.5 and RCP8.5 projected LULCC for the boreal summer (June-July-August) averaged for the final 10 years of simulations. Regions with dots indicate that the changes are significant at the 90% confidence level. These are results from Off-line runs with prescribed meteorology; i.e., meteorological variables do not respond to LULCC.

*14. Line 519. Where do we see that soil is drier?.*

The soil moisture changes have been added to Figure 5.

[Figure]

RCP4.5 Projected LULCC

(a) Surface O$_3$ (ppbv)   (b) Isoprene emission ($10^{-10}$ kg m$^{-2}$ s$^{-1}$)   (c) Dry deposition velocity (cm s$^{-1}$)

(d) Forest coverage (%)   (e) Albedo ($10^{-2}$)   (f) Surface net solar radiation (W m$^{-2}$)

(g) Surface temperature (K)   (h) Precipitation (mm day$^{-1}$)   (i) Soil Moisture top 10 cm (mm)

Figure 5. Changes in surface ozone, isoprene emission, dry deposition velocity, projected forest coverage, simulated surface albedo, surface net solar radiation, surface temperature, precipitation, and soil moisture at top 10-cm layer, during the boreal summer over the US due to RCP4.5 projected LULCC. Regions with dots indicate that the changes are significant at the 90% confidence level.

*15. Line 544. Why not show the same evidence as for Figure 6, perhaps in the supplement?*

The same evidence of Figure 6 can be used for Europe, since both US and Europe regions are analysed as zooming-in regions and they are from the same experiment.

Fig. 6 is included here for the reviewer's information:

[Figure]

Figure 6. Present-day conditions and changes in zonal wind at 200 hPa and geopotential height at 500 hPa during the boreal summer. Changes due to RCP4.5 projected LULCC change are in middle-bottom left panel, while RCP8.5 in middle-bottom right panel. Regions with dots indicate that the changes are significant at the 90% confidence level.

*16. Line 550. Should this be RCP4.5 here? Otherwise this sentence does not make sense.*

We double-checked and the original sentences are correct:

P32, L605-607: "For RCP8.5, reforestation occurs over limited areas of Europe (Figs. 2l); similar changes in the local climate and surface ozone are found, albeit with a relatively weak amplitude compared with their RCP8.5 counterparts."

In western Europe, both RCP4.5 and RCP8.5 are projected to have more forests (Fig. 2g, l). Thus, the LULC initiated changes are similar, but the projected reforestation is more intense in RCP4.5 than RCP8.5, thus the responds are in more substantial in RCP4.5.

*17. Line 555-556. This discussion is unnecessarily speculative as it should be possible to demonstrate whether this mechanism is operating in the model or not.*

We have revised the sentence to avoid confusion as follows:

P32, Line 611-620: "Again, temperature increase tends to occur west of the LULCC (Fig. 8e). **The LULCC-induced lower albedo (Fig. 8c) and higher net surface solar radiation (Fig. 8d) cause more energy to be absorbed by the land surface at high elevations and warm**

**the overlying air accordingly. Again, the rainbelt is displaced northward, likely reflecting perturbed synoptic-scale activities in the region. Consistent with the former feature, the mid-tropospheric anomalous flow is characterized by an anticyclone between 20–30ºN, suppressing rainfall therein (Figs. 8f, 8g). The anomalous anticyclone in turn can lead to more surface net radiation in northern India as a remote effect (Fig. 8d).** Thus, in northern India there is significant surface warming (Fig. 8e) and enhanced surface ozone."

*18. Line 560-562. Would be helpful to refer to Figure 2 here.*

The referred figure has been added, many thanks for reviewer's suggestion!

P33, Line 624-626: "Finally, in China extensive deforestation occurs for RCP4.5 **(Fig. 2f)**. Surface ozone shows a slightly decrease **(Fig. 4a)** that could be caused by biogeochemical effects associated with LULCC instead of biogeophysical effects."

*19. Line 572. How have local responses been separated out here? Wasn't LULC changed everywhere in the model at the same time?*

To be more precise and in response to the reviewer's concern, we have modified the description as follows:

P34, L633-635: "Overall, we find that biogeophysical effects **can have strong impacts on surface ozone through modifying local and remote meteorological conditions such as** surface warming and circulation anomalies initiated by local LULCC (Fig. 1)."

*20. Line 586. Could refer back to Table 2 here.*

The referred table has been added, many thanks for reviewer's suggestion!

P34, L647-650: "Our results show that changes in ozone, near-surface air temperature, and other factors controlling ozone are similar between transient and time-sliced runs in the On-line mode (Fig. 9 and **Table 2**)."

*21. Line 622-623. Briefly explain how this conclusion was reached. Are these the only mechanisms by which the model can respond to changes in LULC?*

We revised the manuscript accordingly:

P37, L696-703: "The mechanisms behind temperature responses to LULCC can be: **first, surface properties (e.g., surface albedo, evapotranspiration, and surface roughness) are altered; the changes in surface properties lead to changes in surface energy budget and surface temperature locally; then these changes would propagate to upper levels of the atmosphere and further induce a regional circulation response, in particular the formation of anomalous stationary high-pressure systems and warming conditions over the mid-to-high northern latitudes in boreal summer as a remote effect.**"

*22. Figure 1. Some additional detail could be added here. For example, in the green box, won't the sign of the change depend on the location dependent to how the westerlies are displaced? Similarly, in the yellow box, should the sign of the change depend on local chemistry and emissions in the case of isoprene, and also on location with respect to the anomalous high and changes in moisture divergence or net surface radiation?*

We have revised the figure by adding more details suggested by the reviewer, including in the green box, "Displacement of westerlies" is revised as "Displacement of upper-/mid-tropospheric flow" indicating the upper level westerlies poleward beyond the range of 20 degree latitude (US and Europe cases), and middle to high tropospheric flows between 15 and 20 latitude (India case). In the yellow box, the isoprene emission and ozone relationship is changed from "+" to "+/-". We also added the dry deposition changes related to humidity changes. Please find our modification for Fig. 1.

[Figure]

Figure 1. Schematic diagram showing the biogeochemical and biogeophysical effects of land use and land cover change (LULCC) on surface ozone, using a case where forest coverage increases (e.g., under the RCP4.5 scenario) as an example. Red arrows indicate biogeochemical effects and grey arrows indicate biogeophysical effects. We focus on processes initiated at the land surface by LULCC changes, and the corresponding responses in local near-surface atmosphere (blue box), middle- to upper-level atmosphere (green box) and remote near-surface atmosphere (yellow box).

*23. Table 2. Why not add columns for the changes in surface ozone here?*

We have added another column for the changes in surface ozone. The global average ozone concentration has been included. Signals between increases and decreases have been merged, resulting in an overall of global ozone concentration a small number, up to 1% change. Thus, some hotspots suffering ozone pollution have been investigated in this manuscript. Please find our modification for Table 2.

*24. Figure 4. How has significance been assessed? This should be explained in the methods section.*

We have added the significance in the method session:

P14 L301-302: "**The statistical significance of the comparison amongst these experiments was assessed by the Student's t-test at the 90% confidence levels**."

We also added the description in each related figure's caption: **"Regions with dots indicate changes that are significant at the 90% confidence level."**

Responses to Referee #2

*Wang et al. aimed to address the impact of future land use and land cover change (LULCC) on surface ozone. They authors differentiated between the role of biogeochemical effects and biogeophysical effects by conducting fixed-dynamics simulations and coupled chemistry-climate simulations using a 3-D global model. They found that the biogeochemical effects are relatively small due to the counteracting impacts of isoprene emission and ozone uptake by plants. The biogeophysical processes likely play a more important role both at local scale through albedo effects and surface energy redistribution, and at regional scale through teleconnections. The manuscript is well structured and clearly written, and fits well into the scope of ACP. Below are some minor issues to be considered before publication.*

We thank the reviewer for the comments.

*Throughout the manuscript, I do not see a discussion on whether the changes in ozone are significant, compared to the perturbations caused by natural climate variability. E.g. Fig 3 needs to include a confidence level. I see some dots in other figures but there are no explanations.*

We have added the significance in the method session:

P14 L301-302: "**The statistical significance of the comparison amongst these experiments was assessed by the Student's t-test at the 90% confidence levels**."

We also added the description in each related figure's caption: **"Regions with dots indicate changes that are significant at the 90% confidence level."**

We also added a paragraph in the discussion section on the comparison between interannual variability and LULCC-induced climate signals:

P39-40, L747-769: "**Atmospheric internal variability is one factor that could affect the significance of our results. Large internal variability of the climate system reduces the signal-to-noise ratio for LULCC-induced climatic changes. This is analogous to the problem of long-term low-frequency variability of the extratropical circulation affecting the interpretation and extraction of climate change signals, especially if short time series (e.g., ~10 years) are used (Deser et al., 2012). When applicable, multiple-member ensemble runs are required to ascertain the impacts of such variability. Our model experiments with a 55-year transient integration with prescribed sea surface temperature and sea ice are not designed to address such low-frequency climate variability. We note however that our climate simulations focus on land-atmosphere biogeophysical interactions, which typically operate on a shorter timescale, and thus the LULCC-induced climate signals that we detected are expected be present when superimposed upon any long-term trajectory and low-frequency variability undergone by the climate system. For land-atmosphere interactions, high-frequency interannual variability on a decadal timescale may be more relevant. To assess the potential impacts of the internal variability of the system on a decadal timescale, we have compared the magnitudes of interannual standard deviations of near-surface temperature of the CTL run with the LULCC-induced climate signals. Our results show that the climate signals are not weak and can be comparable to interannual variability at midlatitudes (Fig. S4), e.g., over North America land areas at ~45°N and also north of 60°N. It is also noteworthy that the time-sliced experiments with single-year forcing looped for multiple years, give results very similar to the transient simulations, pointing to the robustness of LULCC impacts.**"

[Figure]

Figure S4. Standard deviations of near-surface air temperature in CTL run (a), compared with changes induced by LULCC from On-line runs under two future scenarios: RCP4.5 (b) and RCP8.5 (c) during the boreal summer.

*The authors may also add some explicit discussion on ozone changes induced by anthropogenic emissions under RCP4.5 and RCP8.5, to give the readers a clearer idea what the LULCC impacts are compared to emissions and climate.?*

*L432: Give a rough number/range of ozone responses to changes in anthropogenic emissions and climate here.*

We have added discussion as follows:

P36-37, L681-695: "The surface ozone changes due to future LULCC are comparable with anthropogenic emissions and climate, and thus should be taken into account in future research and policy planning. **For example, summertime surface ozone changes induced by climate change alone are projected to increase by 1–10 ppb in the US, Europe, East and South Asia (e.g., Jacob and Winner, 2009; Fiore et al., 2012). It is also found that the combined effects of changing climate, emissions and land cover on surface ozone are up to 10 ppb in the US under two RCP scenarios, and the contributions from the three factors have comparable magnitudes although of different signs (Val Martin, et al., 2015). Wang et al. (2011) found that in China, summertime surface ozone decreases by ~10 ppb on average with a maximum reduction of 25 ppb if all anthropogenic emissions are removed. Our simulated ozone changes induced by LULCC are substantial and within the same order of magnitude as the above studies and others that considered meteorological responses to LULCC (Ganzeveld et al., 2010; Val Martin et al., 2015). This highlights the important roles of LULCC in modulating surface ozone.**"

*L130, L131 and many other places: LULC should rather be land use and land cover change (LULCC). Land cover itself cannot 'induce' anything. I'd suggest to check through the manuscript and use LULCC instead of LULC.*

We have revised the term throughout the manuscript.

*L221: CAM4 should be "On-line mode of CAM4-Chem-CLM".*

We have revised accordingly.

P11, L241-243: "**For the On-line mode of CAM4-Chem-CLM, 26 vertical levels are used between the surface and 4 hPa, with the same horizontal resolution as the Off-line mode**."

*L225: I think you prescribed anthropogenic emissions, biomass burning and long-lived species CO2, CH4 and N2O for ALL simulations, not just the 5 coupled simulations, right? Combine L224-225 with L212-213 together and provide more details, e.g. what inventories/values you used.*

The reviewer is right. We have revised accordingly.

P11, L243-245: "**For all simulations** , concentrations of long-lived greenhouse gases including CO2, CH4, and N2O are prescribed at present-day levels ."

The inventory has been included:

P11, L245-246: "**For the anthropogenic emissions used for all simulation are described in Lamarque et al. (2010, 2012) and references therein.**"

*Fig. 3g, k, h, l: these plots are not clear. Perhaps change the scale of color bars.*

Figure 3 has been revised with larger view, thus the changes in variables should look better now. For better comparison, we keep the color scale in Figure 3 the same as that in Figure 4.

*L408-409: Shouldn't the tropical region (e.g. the Amazonia) be the same regime as the southeast US? With increases in isoprene emission that consumes more ozone and increases in dry deposition, I assume ozone will decrease as in the southeast US. This is not shown in Fig 3g, why? Again, change the color bar may improve the visualization, and adding a map showing relative changes in percentage in the supplement would help.*

Southeast US and tropical regions share the same regime of ozone formation, as mentioned by the reviewer. In Fig. 3g, ozone decreases slightly over the southeast US due to the forest increases locally (Fig. 2f). Slight decrease of ozone is also found in the tropical regions (e.g., the Amazonia and Equatorial Africa). More details can be seen from the relative changes in percentage in the supplementary, and from the revised text.

P23, L442-443: "**Suppressed ozone is also found in the tropical regions of South America and Africa (Fig. S1a)**."

*L452-458: This paragraph is not clear. It reads like the changes in isoprene emission and dry deposition are due to meteorological changes (warmer temperature, drier/wetter conditions), but actually you cannot differentiate whether the changes are directly caused by LULCC or LULCC-induced meteorological changes, right? You mentioned that isoprene emission changes are smaller than the off-line values, suggesting meteorology partly offsets the direct LULCC impacts. But intuitively, reforestation leads to warming (mid-latitudes in Fig 4d) and then more isoprene emission,*

*so the LULCC- induced meteorology changes add on to the direct LULCC impact on isoprene?*

To avoid unclear expression, we revised the paragraph as follows:

P25, L483-487: "Changes in isoprene emission also correlate with temperature changes (Figs. 4b, d; Figs. 4f, h, Figs. 5a, b, Fig. S1b). Isoprene emission also increases in regions with forest expansion, **reflecting not only the biogeochemical effects due to higher fractional coverage of isoprene-emitting vegetation types (Section 3.2), but also the biogeophysical effects arising from changing land surface temperature.**"

P25-26, L502-506: "Table 2 shows **in general, the percentage changes in isoprene emission and dry deposition in the On-line simulations are smaller than in the Off-line simulations in both scenarios, reflecting that on a global scale, LULCC-induced meteorological changes partly offset the biogeochemical effects of changing land cover types on ozone.**"

*L462-463: Avoid using the terms "biogeophysical" and "biogeochemical" here. You are referring to impacts on isoprene emission, not ozone.*

We have revised the related words.

*L508-514: How will the anomalous high with low-level divergent wind in RCP4.5 experiment influence ozone advection and transport?*

We have added:

P29, L568-571: "**The low-level divergent wind, on the other hand, can also prevent ozone and its precursors along the West Coast from being advected eastward due to the blocking from the Rocky Mountains, resulting in enhanced ozone pollution over the western US.**"

*L538: What's the meaning of "a positive tendency" for the energy budget?*

We revised as:

P31, L592-594: "Considering all surface energy components,  a positive net heat flux and thus a surface temperature increase are found (Fig. 7e)."

*L576-578: Also add Unger 2014 Nat. Clim. here for the cooling of deforestation.*

We added the reference of Unger 2014 in the manuscript:

P34, L639-641: "In addition, Govindasamy and Caldeira (2001) and **Unger (2014)** also found surface cooling due to deforestation."

Responses to Referee #3

*This paper looks at the biogeochemical and biogeophysical impacts of land-cover change on surface ozone. Surface ozone perturbations are much larger when including the biogeophysical changes, than when looking at the biogeochemical aspects alone. This is an interesting result and could be published.*

*However, the paper suffers due to methodological and conceptual errors. Considering the internal variability of the atmosphere, the authors need to do considerably more work to show that the biogeophysical ozone signal is due to changes in the land cover instead of internal variability. This paper might be publishable after extensive revisions.*

> We thank the reviewer for the comments. The natural variability is revised, pls find more details as follows.

*1. I believe the authors may have looked at the statistical significance of their difference maps (the figures appear to show some cross-hatching), but they are difficult to read and are not mentioned in the text. Statistically significant areas need to be highlighted and the significance level discussed in the text. In many cases non-significant signals are discussed. They should not be. For example, there is extensive discussion of the ozone signal in Europe, but judging from Figure 4 these changes are not significant.*

> We have added the significance in the method session:

> P14 L301-302: "**The statistical significance of the comparison amongst these experiments was assessed by the Student's t-test at the 90% confidence levels**."

> We also added the description in each related figure's caption: **"Regions with dots indicate changes that are significant at the 90% confidence level."**

> We also added a paragraph in the discussion section on the comparison between interannual variability and LULCC-induced climate signals:

> P39-40, L747-769: "**Atmospheric internal variability is one factor that could affect the significance of our results. Large internal variability of the climate system reduces the signal-to-noise ratio for LULCC-induced climatic changes. This is analogous to the problem of long-term low-frequency variability of the extratropical circulation affecting the interpretation and extraction of climate change signals, especially if short time series (e.g., ~10 years) are used (Deser et al., 2012). When applicable, multiple-member ensemble runs are required to ascertain the impacts of such variability. Our model experiments with a 55-year transient integration with prescribed sea surface temperature and sea ice are not designed to address such low-frequency climate variability. We note however that our climate simulations focus on land-atmosphere biogeophysical interactions, which typically operate on a shorter timescale, and thus the LULCC-induced climate signals that we detected are expected be present when superimposed upon any long-term trajectory and low-frequency variability undergone by the climate system. For land-atmosphere interactions, high-frequency interannual variability on a decadal timescale may be more relevant. To assess the potential impacts of the internal variability of the system on a decadal timescale, we have compared the magnitudes of interannual standard deviations of near-surface temperature of the CTL run with the LULCC-induced climate signals. Our results show that the climate signals are not weak and can be comparable to interannual variability at midlatitudes (Fig. S4), e.g., over North America land areas at ~45°N and also north of 60°N. It is also noteworthy that the time-sliced experiments with single-year forcing looped for multiple**

**years, give results very similar to the transient simulations, pointing to the robustness of LULCC impacts.**"

CTL Standard Deviation over the last 10 yrs

(a) Temperature (degree)

[Figure]

RCP4.5 projected 2050 LULC changes    RCP8.5 projected 2050 LULC changes

(b) Temperature (degree)          (c) Temperature (degree)

[Figure]

[Figure]

Figure S4. Standard deviations of near-surface air temperature in CTL run (a), compared with changes induced by LULCC from On-line runs under two future scenarios: RCP4.5 (b) and RCP8.5 (c) during the boreal summer.

For Europe, significant ozone changes are found along coastal areas. Since western Europe is one of the most polluted regions due to ozone, the changes along its coastal areas also have critical implications to the local air quality. To be more precise, we revised the manuscript as follows:

P30, L586-588: "**Along coastal areas of Europe**, substantial increases in surface ozone (Figs. 4a, e) and near-surface air temperature (Figs. 4d, h) are found due to RCP4.5 and RCP8.5 LULCC."

*2. The response of the atmosphere to the land-surface is complex. It is not as simple as simply applying the thermal wind balance to surface temperature perturbations. If the authors do wish to include the cause and effect of the atmospheric perturbations to land-cover change they would do well to enlist the help of an atmospheric dynamicist. As it stands he paper will only be strengthened by omitting the rather simplistic meteorological explanations of the impact of land-cover change on the atmosphere.*

*There have been many simulations of the atmosphere to perturbations of surface temperatures (noting the response is much different in the tropics than the mid-latitudes). The authors could support their hypothesis by citing relevant papers. On the other hand, many studies do show a response in the general circulation to changes in land-cover (e.g., see Lague et al. [2019, preprint DOI,*

*10.31223/osf.io/dbyqu] and references therein.) I am struck by the very similar northwards displacement of the jet-stream in the RCP4.5 and RCP8.5 simulations, despite different land cover changes and changes in the surface temperature response. This may argue for similar changes in the overall circulation. These changes appear to be on the hemispheric scale. It is unclear if any of the local changes are really significant.*

We added one paragraph in Introduction about the complex impact of LULCC on atmosphere:

P6-7, L120-141: "LULCC can also affect weather and climate over local and remote regions by perturbing the biosphere-atmosphere exchange of water and energy fluxes (e.g., Betts, 2001; Bonan, 2016; Pitman et al., 2009). **For example, afforestation generally cools the surface in tropical regions, where evaporative cooling generally exceeds radiative warming from reduced albedo, but warms the surface in boreal forests due to the more dominant radiative warming effect (e.g., Arora and Montenegro, 2011; Lee et al., 2011; Bonan, 2008). There is little consensus on the effects of afforestation in midlatitude regions (e.g., Boisier et al., 2012; de Noblet-Ducoudré et al., 2012). Furthermore, the impacts of such surface forcing could extend into the upper troposphere, alter large-scale circulation pattern, and consequently affect the climate in remote regions (Henderson-Sellers et al. 1993; Chase et al., 2000; Swann et al., 2012). Recent studies (Devaraju et al. 2015; Laguë and Swann 2016) have identified that LULCC in midlatitude regions can modify the global energy balance, impacting cloud cover, precipitation, and circulation pattern via remote effects. By and large, the impacts of LULCC on the atmosphere is complex. Laguë et al. (2019) examined the climatic effects of individual physical components in the land surface (albedo, evaporative resistance and surface roughness), and found that temperature responds most to changes in albedo and evaporative resistance, particularly in the extra-tropics through large-scale atmospheric feedbacks. Still, how individual land characteristics play out together and interact with each other to affect the atmospheric general circulation, and how the surface signals translate into those in the upper levels are not fully understood.**"

We also have added several representative studies to support cause and effect of the atmospheric perturbations to LULCC (e.g. Chase et al., 2000; Swann et al., 2012; Medvigy, et al., 2013; Xu et al., 2015). We have revised the global scale thermal wind description into a more local-scale storm track and associated jet stream displacement driven by temperature gradient change (between mid-latitude and polar area, between land and ocean). The revision is as follows:

P28, L546-557: "The warming over these regions is likely related to atmospheric circulation changes over the northeastern US. **Many studies have found that LULCC-induced surface changes can propagate to upper levels vertically and to higher latitudes meridionally (e.g., Chase et al., 2000; Swann et al., 2012; Medvigy, et al., 2013; Xu et al., 2015), resulting in remote effects of LULCC. In our study,** surface warming in relation to reduced albedo over the northeastern US **can lead to the upper-level warming up to 200 hPa (not shown here). This warming at midlatitudes can lead to anomalous meridional temperature gradient, resulting in the storm track as well as the westerly jet at midlatitudes being displaced northward.** Inspection of the anomalous zonal wind at 200 hPa indicates that the westlerly jet core is displaced northward from its climatological position at ~50°N (see Figs. 6a and 6c)."

P32, L611-620: "Again, temperature increase tends to occur west of the LULCC (Fig. 8d). **The LULCC-induced lower albedo (Fig. 8c) and higher net surface solar radiation (Fig. 8d) cause more energy to be absorbed by the land surface at high elevations and warm the overlying air accordingly. Again, the rainbelt is displaced northward, likely reflecting perturbed synoptic-scale activities in the region. Consistent with the former**

**feature, the mid-tropospheric anomalous flow is characterized by an anticyclone between 20–30ºN, suppressing rainfall therein (Figs. 8f, 8g). The anomalous anticyclone in turn can lead to more surface net radiation in northern India as a remote effect (Fig. 8c).** Thus, in northern India there is significant surface warming (Fig. 8e) and enhanced surface ozone."

For the similar jet-stream northward displacement in RCP4.5 and RCP8.5, as the reviewer mentioned, we also added accordingly as follows:

P38, L717-722: "**A similar northward displacement of the jet stream in the RCP4.5 and RCP8.5 simulations is also found, despite quite different LULCC patterns on a regional scale; this indicates that the mostly extratropical afforestation in RCP4.5 vs. the mostly tropical deforestation in RCP8.5 can lead to similar hemispheric-scale circulation changes that likely reflects common connections to the warming at midlatitudes.**"

*3. In their interpretation of the response to landcover change the authors should be mindful of the large internal variability of the atmosphere. Even where the differences are found to be statistically significant the interpretation of these differences to changes in land-cover (instead of internal variability) may be problematical. The differences in the simulations could be simply due to decadal variability in the atmosphere. As shown in Deser et al. (2012) [Clim Dyn (2012) 38:527–546DOI 10.1007/s00382-010-0977-x], for example, in most places it takes more than 30 ensembles of 10 year average differences in transient simulations (e.g., 2028–2037minus 2005–2014) to see significant differences in precipitation. While the present simulations might have less variability due to the fixed sea-surface temperatures it is unclear to me how much this reduces the variability. The presence of internal variability may obfuscate any signal from the change in land surface. For example, if I take the stipples in Figure 6 as grid points with significant differences (?) there are many regions of stipples throughout the world (even in parts of the S.H.) which seem significant. If one takes a significance level of 95%, this suggests 5% of the points may only appear to be statistical different.*

*However, the timeslice experiments in essence add another ensemble member. Similarities between the timeslice analysis and the transient analysis may point to robust differences. The authors need to do more work to attribute the changes to changes in land-cover. Their meteorological attribution, as described above, is probably not correct.*

We added one paragraph discussing more extensively the issues arising from internal variability of climate. Please see our responses to Major Comment #1 above.

*Minor Changes*

*1. I felt the paper could be better referenced. Please back up with more references, e.g. L106-108 "Dry deposition. . .." L113-114 "The dry deposition. . . .. L236-237and other locations. . ..*

We added the reference in the mentioned place, and also added other reference to other places in the manuscript:

P5, L107-108: "Dry deposition is another key factor modulating ozone **(e.g., Wesely, 1989; Val Martin et al., 2014; Lin et al., 2019)**."

P6, L114-117: "The dry deposition enhancement mostly arises from climate- and CO2-induced increase in leaf area index (LAI), which more than offsets the compensating effect of cropland expansion **(Fu and Tai, 2015)**."

P12, L256-259: "Uncertain emissions, coarse resolution (Lamarque et al., 2012), misrepresentation of dry deposition process (Val Martin et al., 2014) and overestimation of stomatal resistance **(Lin et al., 2019)** are all likely factors contributing to these high biases."

*2. It is probably important to emphasize somewhere that the impact of the surface on the atmosphere is complex. For example, taking Lague et al. (2019, preprint DOI, 10.31223/osf.io/dbyqu) as an example, the impact can be through changes in albedo, evaporative resistance, and surface roughness. How these play out together and interact with each other is not simple. These changes may impact the clouds, boundary layer turbulence etc. A short paragraph explaining these influences might be in order in the introduction. I don't think the paper mentions surface roughness anywhere. In addition, the explicit impact of the surface on the boundary layer should be discussed as this will impact the dry deposition of ozone and the mixing and venting of ozone in the boundary layer.*

We have modified the introduction and discussion substantially to address these useful points raised by the reviewer. Please first see our responses to Major Comment 2 above about the complex impacts of the surface on the atmosphere.

We also added one more paragraph in Results to include the roughness length and its impacts on dry deposition:

P25, L495-501**: "Furthermore, changes in surface roughness can influence aerodynamic resistance and thus dry deposition via modifying boundary-layer turbulence. In LULCC scenarios, surface roughness is modified substantially with increases in RCP4.5 (Fig. 2j) and reductions in RCP8.5 (Fig. 2o), which generally decrease (increase) resistance and enhance (decrease) dry deposition in RCP4.5 (RCP8.5), though the overall changes in dry deposition is more dominantly shaped by the integrated meteorological effects of LULCC."**

*3. Figure 1. While the land surface may influence the upper troposphere, the exact connection is not really clear. In addition, the tropical response (where there is no real jet-stream) is likely much different than the mid-latitude response.*

We have revised Fig. 1 accordingly.

*4. Data and Methods Section. Prior to going into the model specifics, it might be useful to give a broad overview of the simulations. For example, it was confusing when the paper first discussed online and offline simulations and the setup of both. Note also the online and offline simulations likely will have very different boundary layers with different clouds and radiation so a comparison of these two model setups is not straightforward (e.g., Brownsteiner et al., 2015).*

We have added an overview of the simulations:

P11, L232-237: "**These two modes are both applied in the study. In particular, the Off-line mode is used to quantify the biogeochemical effects of LULCC alone on surface ozone in the absence of any associated meteorological responses to LULCC. The On-line mode is applied to assess the biogeophysical and integrated effects on ozone caused by LULCC, considering also the effects of the resulting meteorological changes.**"

We also revised the model setups to avoid comparison between online and offline simulations, and the revision is as follows:

P11, L238-243: "**For the Off-line mode, we use the Goddard Earth Observing System Model Version 5 (GEOS-5) (https://rda.ucar.edu/datasets/ds313.0/) (Tilmes, 2016)**

**assimilated meteorology as the driving fields, with a horizontal resolution of 1.9°×2.5° and 56 vertical levels between the surface and the 4-hPa level. For the On-line mode of CAM4-Chem-CLM, 26 vertical levels are used between the surface and 4 hPa, with the same horizontal resolution as the Off-line mode.**"

*5. Could the authors clarify the difference in dry-deposition in the off-line land cover change simulations? Are the differences shown only due to differences in stomatal conductance. Is the dry deposition also sensitive to LAI or the type of vegetation even without considering stomatal conductance? I assume the parameterized boundary layer turbulence is the same in each case, correct?*

We have added one paragraph illustrating the dry deposition changes in On-line and Off-line runs:

P25, L489-501: **"In the dry deposition scheme, stomatal resistance can respond to atmospheric dryness and soil water stress. For instance, drier conditions are captured in RCP4.5 in the central-western US as initiated by the LULCC further east, with anomalously low precipitation (Fig. 5h) and soil moisture (Fig. 5i). The drier conditions could result in suppressed dry deposition in the corresponding regions (Fig. 5c). The responses of dry deposition to drought conditions have also been observed by recent studies (e.g., Lin et al., 2019). Furthermore, changes in surface roughness can influence aerodynamic resistance and thus dry deposition via modifying boundary-layer turbulence. In LULCC scenarios, surface roughness is modified substantially with increases in RCP4.5 (Fig. 2j) and reductions in RCP8.5 (Fig. 2o), which generally decrease (increase) resistance and enhance (decrease) dry deposition in RCP4.5 (RCP8.5), though the overall changes in dry deposition is more dominantly shaped by the integrated meteorological effects of LULCC.**

Table 2 shows **in general, the percentage changes in isoprene emission and dry deposition in the On-line simulations are smaller than in the Off-line simulations in both scenarios, reflecting that on a global scale, LULCC-induced meteorological changes partly offset the biogeochemical effects of changing land cover types on ozone**."

*6. L641 "vice versa", please spell out.*

We have revised the text as follows:

P18, L367-368: "Forest expansion leads to increases in LAI, **whereas deforestation results in LAI reductions** "

*7. L357, Do the isoprene emissions depend on the makeup of the forest expansion?.*

Yes. We have added one sentence to clarify:

P10, L210-218: "In CLM4.5, biogenic VOC emissions are computed using the Model of Emissions of Gases and Aerosols from Nature (MEGAN) version 2.1 (Guenther et al., 196 2012), accounting for the major known processes controlling biogenic VOC emissions from terrestrial ecosystems, such as effects of temperature, solar radiation, soil moisture, leaf age, $CO_2$ concentrations, and vegetation species and density. Biogenic VOC emissions in MEGAN are allowed to respond interactively to changes of these processes. **Thus, isoprene emission is allowed to respond to spatiotemporal changes in PFTs and the associated changes in meteorological conditions in this study.**"

In the off-line configurations, meteorological conditions remain unchanged, so isoprene emissions can respond to changes of surface properties, and here the only changes are from

the PFT type changes and associated emission changes.

*8. It is unclear why NOX is shown in Figure 3. I don't think it is discussed.*

We remove the $NO_x$ figure. Please refer to the new figure 3.

*9. In discussing the biogeophysical response the authors did not discuss changes in the ozone deposition velocity. This seems to be a field that would be easy to show and could be more directly attributable to land cover change.*

We have added one paragraph discussing the changes in ozone deposition velocity. Please refer to our responses to Minor Comment 5.

*10. Figure 4 and other figures showing the difference between online model simulations. I believe these figures show the difference between the final 10 years of the transient simulation and 10 years from the on-line CTL. Please state explicitly (maybe in the figure caption?).*

We have added the information in captions of Fig. 4 and Fig. 3 as follows:

"Figure 3. Simulated 2000-to-2050 changes in surface ozone, isoprene emission, dry deposition velocity and surface NOx under RCP4.5 (middle) and RCP8.5 projected LULCC for the boreal summer (June-July-August) **averaged for the final 10 years of simulations**."

"Figure 4. Simulated 2000-to-2050 changes in surface ozone, isoprene emission, dry deposition velocity and near-surface air temperature with atmosphere-chemistry-land coupled configurations for the boreal summer **averaged over the 10-year analysis window**, under two future scenarios (RCP4.5 and RCP8.5)."

*11. In discussing figure 4, the authors mention the correlations between difference fields. In each case please give the correlation coefficient between the fields and its significance. I would suspect that in many cases these correlations are not actually significant, in which case the authors need to refine their language in discussing the relation between the fields.*

We have added the figure including the correlation coefficients, and revised the text accordingly, for example:

P24, L472-476, L483-484: "**On the other hand, changes in surface temperature are found to be correlated well with patterns of changes in ozone (Fig. S2a, d), indicating that the biogeophysical drivers that modify temperature may play critical roles in ozone changes. In the regions where temperature increases, surface ozone increases correspondingly.**"… "**Changes in isoprene emission also correspond closely temperature changes (Figs. 4b, d; Figs. 4f, h, Fig. S2b).**"

*12. L457 "meteorological changes", these would include not only stomatal response which is part of the story, but the impacts of surface roughness and surface heat exchange on boundary layer turbulence.*

We have added more explanation. Please refer to our responses to Minor Comment 5 above.

*13. L502 -L520 (also L542-L550). Please delete. This is rather speculative. The response to land forcing is likely to be complex. The argument concerning the thermal wind relation and the jet-stream is pretty "hand-wavey" and I doubt it is correct. Has anyone else seen this? The changes are most likely dynamically consistent with each other (as described in the paragraph), but this is much different than arguing that they are due to changes in the land surface and in particular through the mechanism described.*

Please refer our responses to Major Comment 2 above.

*14. L522 What is the correlation coefficient?*

We have added the correlation coefficient figures in supplementary, and cited the figures accordingly.

*15. Fig. 5 and Fig. 7 and Fig 8. The discussion would be clearer if you put in a panel showing surface ozone (I don't think you need to show the topography in Fig. 5; in Fig. 8 most of the changes don't appear significant).*

We have added the ozone concentration, and revised the figure accordingly.

*16. L555-556, "suggesting"... This is doubtful. India is in the subtropics. The atmospheric response to land cover change is likely to be rather different than in the midlatitudes.*

We have revised the explanation about India extensively. Please see our responses to Major Comment 2 above.

**References:**

Guenther, A. B., Jiang, X., Heald, C. L., Sakulyanontvittaya, T., Duhl, T., Emmons, L. K., and Wang, X.: The Model of Emissions of Gases and Aerosols from Nature version 2.1 (MEGAN2.1): an extended and updated framework for modeling biogenic emissions, Geosci. Model Dev., 5, 1471-1492, https://doi.org/10.5194/gmd-5-1471-2012, 2012.

Wesely, M.: Parameterization of surface resistances to gaseous dry deposition in regional-scale numerical models, Atmos. Environ., 23, 1293-1304, https://doi.org/10.1016/0004-6981(89)90153-4, 1989.

Emmons, L. K., Walters, S., Hess, P. G., Lamarque, J.-F., Pfister, G. G., Fillmore, D., Granier, C., Guenther, A., Kinnison, D., Laepple, T., Orlando, J., Tie, X., Tyndall, G., Wiedinmyer, C., Baughcum, S. L., and Kloster, S.: Description and evaluation of the Model for Ozone and Related chemical Tracers, version 4 (MOZART-4), Geosci. Model Dev., 3, 43-67, https://doi.org/10.5194/gmd-3-43-2010, 2010.

Lamarque, J. F., Emmons, L. K., Hess, P. G., Kinnison, D. E., Tilmes, S., Vitt, F., Heald, C. L., Holland, E. A., Lauritzen, P. H., Neu, J., Orlando, J. J., Rasch, P. J., and Tyndall, G. K.: CAM-chem: description and evaluation of interactive atmospheric chemistry in the Community Earth System Model, Geosci. Model Dev., 5, 369-411, https://doi.org/10.5194/gmd-5-369-2012, 2012.

Val Martin, M., Heald, C. L., and Arnold, S. R.: Coupling dry deposition to vegetation phenology in the Community Earth System Model: Implications for the simulation of surface $O_3$, Geophys. Res. Lett., 41, 2988-2996, https://doi.org/10.1002/2014GL059651, 2014.

Taylor, K. E., Stouffer, R. J., and Meehl, G. A.: An overview of CMIP5 and the experiment design, Bull. Am. Meteorol. Soc., 93, 485-498, https://doi.org/10.1175/BAMS-D-11-00094.1, 2012.

Hurtt, G. C., Chini, L. P., Frolking, S., Betts, R. A., Feddema, J., Fischer, G., Fisk, J. P., Hibbard, K., Houghton, R. A., Janetos, A., Jones, C. D., Kindermann, G., Kinoshita, T., Goldewijk, K. K., Riahi, K., Shevliakova, E., Smith, S., Stehfest, E., Thomson, A., Thornton, P., van Vuuren, D. P., and Wang, Y. P.: Harmonization of land-use scenarios for the period 1500–2100: 600 years of global gridded annual land-use transitions, wood harvest, and resulting secondary lands, Climatic Change, 109, 117-161, DOI:10.1007/s10584-011-0153-2, 2011.

Heald, C. L., and Spracklen, D. V.: Land use change impacts on air quality and climate, Chem. Rev., 115, 4476-4496, https://doi.org/10.1021/cr500446g, 2015.

Heald C. L., Henze, D. K., Horowitz, L. W., Feddema, J., Lamarque, J.‐F., Guenther, A., Hess, P. G., Vitt, F., Seinfeld, J. F., Goldstein, A. H., and Fung, I.: Predicted change in global secondary organic aerosol concentrations in response to future climate, emissions, and land use change, J. Geophys. Res., 113, D05211, doi:10.1029/2007JD009092, 2008.

Wu, S., Mickley, L. J., Kaplan, J. O., and Jacob, D. J.: Impacts of changes in land use and land cover on atmospheric chemistry and air quality over the 21st century, Atmos. Chem. Phys., 12, 1597-1609, https://doi.org/10.5194/acp-12-1597-2012, 2012.

Lelieveld, J., Butler, T. M., Crowley, J. N., and co-authors: Atmospheric oxidation capacity sustained by a tropical forest. Nature, 452, 737-740, doi:10.1038/nature06870, 2008.

Arora, V. K., and Montenegro, A. Small temperature benefits provided by realistic afforestation efforts. Nat. Geosci., 4, 514-518. https://doi.org/10.1038/ngeo1182, 2011.

Lee, X., Goulden, M. L., Hollinger, D. Y., Barr, A., Black, T. A., Bohrer, G., …Zhao, L. Observed

increase in local cooling effect of deforestation at higher latitudes. Nature, 479(7373), 384–387. https://doi.org/10.1038/nature10588, 2011.

Bonan, G. B. Forests and climate change: Forcings, feedbacks, and the climate benefits of forests. Science 320(5882), 1444–1449. DOI: 10.1126/science.1155121, 2008.

Boisier, J. P., de Noblet-Ducoudré, N., Pitman, A. J., Cruz, F. T., Delire, C., van den Hurk, B. J. J. M., . . . Voldoire, A. (2012). Attributing the impacts of land-cover changes in temperate regions on surface temperature and heat fluxes to specific causes: Results from the first LUCID set of simulations. Journal of Geophysical Research: Atmospheres, 117, D12. https://doi.org/10.1029/2011JD017106

de Noblet-Ducoudré, N., Boisier, J. P., Pitman, A., Bonan, G. B., Brovkin, V., Cruz, F., …Voldoire, A. (2012). Determining robust impacts of land-use-induced land cover changes on surface climate over North America and Eurasia: Results from the first set of LUCID experiments. Journal of Climate, 25, 3261–3281. https://doi.org/10.1175/JCLI-D-11-00338.1

Deser, C., Knutti, R., Solomon, S., Phillips, A.: Communication of the role of natural variability in future North American Climate. Nature Clim. Change, 2, 775-779, doi:10.1038/nclimate1562, 2012.

Devaraju, N., Bala, G., Modak, A. Effects of large-scale deforestation on precipitation in the monsoon regions: Remote versus local effects. PNAS, 112, 3257-3262, doi:10.1073/pnas.1423439112, 2015.

Laguë, M., and Swann, A. S. Progressive midlatitude afforestation: Impacts on clouds, global energy transport, and precipitation. J. Clim., 29, 5561-5573, doi: 10.1175/JCLI-D-15-0748.1, 2016.

Laguë, M. M., Bonan, G. B., Swann, A. S. Separating the impact of individual land surface properties on the terrestrial surface energy budget in both the coupled and un-coupled land-atmosphere system. J. Clim. Preprint. 2019

Thornton, J. A., Wooldridge, P. J., Cohen, R. C., Martinez, M., Harder, H., Brune, W. H., Williams, E. J., Roberts, J. M., Fehsenfeld, F. C., Hall, S. R., Shetter, R. E., Wert, B. P., and Fried, A.: Ozone production rates as a function of NOx abundances and HOx production rates in the Nashville urban plume, J. Geophys. Res., 107, 4146(D12), 4146, doi:10.1029/2001JD000932, 2002.

Kubistin, D., Harder, H., Martinez, M., Rudolf, M., … , and Lelieveld, J. Hydroxyl radicals in the tropical troposphere over the Suriname rainforest: comparison of measurements with the box model MECCA. Atmos. Chem. Phys., 10, 9705-9728, 2010. doi:10.5194/acp-10-9705-2010.

---

## Author Response (AR2)

Author Responses to Referees' Comments on **"Impacts of future land use and land cover change on mid-21st-century surface ozone air quality: Distinguishing between the biogeophysical and biogeochemical effects"** by L. Wang et al. (MS No.: acp-2019-824)

Our point-by-point responses are provided below. The referees' comments are *italicized*, our new/modified text is highlighted in **bold**. The revised manuscript with tracked changes is also included in the linked file below for the Editor's easy reference:

https://www.dropbox.com/s/ve8g6isi0p7ttuf/acp-2019-824-manuscript-July2020_trackchanges.docx?dl=0

Responses to Referee #3

*The authors have made substantial changes to the manuscript and it is much improved. It is a potentially very interesting study but as described below I still believe major revisions are necessary prior to publication. I am still concerned about the overall significance and interpretation of their results. The authors have partially addressed the problem of variability in their discussion and conclusions. However, as explained below I am not still convinced of the current results due to the comparatively **weak statistical tests**, **the lack of long averaging times** and the **lack of a convincing dynamical argument**. I have tried to include some specific recommendations.*

> We thank the reviewer for the very helpful comments. More simulations and analysis have been conducted and the paper has been revised substantially to address the reviewer's concerns point by point, and all changes are cited and discussed in the responses below.

*(1) For the paper to be valid the authors need to address the variability of the atmosphere up front, not as an afterthought in the discussion and conclusions. They need to establish beyond a reasonable doubt that the simulation differences are due to changes in LULCC. The problem is not, as stated in the paper (p39, l 756-760) the short timescale of the LULCC circulation changes but that these changes are relatively small (Brovkin et al., 2013) and therefore can be difficult to distinguish from atmospheric noise.*

> We now add these points to address the reviewer's comments:

> First, one paragraph has been added in the Introduction section to convey a reasonable doubt on the large-scale climatic impacts of LULCC:

> P7, Line 142-151: "**By and large, the impacts of LULCC on weather and climate are complex. There is high confidence that LULCC can affect regional climate and climate in remote areas as far as few hundreds of kilometers away (Jia, et al., 2019). The magnitude and sign of regional climate change vary across regions depending on the magnitude of LULCC and background climatic conditions. However, on the global scale, the net changes resulting from LULCC alone are relatively small (e.g., Matthews et al. 2004; Pongratz et al. 2010; Brovkin et al., 2013; Shevliakova et al. 2013; Simmons and Matthews, 2016). Thus, sometimes climatic responses to LULCC may be difficult to distinguish from natural climate variability especially on the global scale.**"

> Discussions on internal variability of climate and LULCC signals have been revised. In light of new simulated results and rearrangement of the presentation order of the time-sliced and transient simulations, redundant sentences have now been removed, and some discussions on the ensemble results have been added:

P39, Line 749-752: "~~This is analogous to the problem of long-term low-frequency variability of the extratropical circulation affecting the interpretation and extraction of climate change signals, especially if short time series (e.g., ~10 years) are used (Deser et al., 2012).~~"

P39, Line 753-761: "~~Our model experiments with a 55-year transient integration with prescribed sea surface temperature and sea ice are not designed to address such low-frequency climate variability. We note however that our climate simulations focus on land-atmosphere biogeophysical interactions, which typically operate on a shorter timescale, and thus the LULCC-induced climate signals that we detected are expected be present when superimposed upon any long-term trajectory and low-frequency variability undergone by the climate system. For land-atmosphere interactions, high-frequency interannual variability on a decadal timescale may be more relevant.~~"

P38-39, Line 764-771: "**… To ascertain the impacts of such variability, we have adopted an analysis period of 30 years for both the time-sliced simulations (looping over the single-year LULCC forcing) and 2-member ensemble transient LULCC simulations. Results from both simulation approaches all show broadly consistent signals induced by LULCC in North America and Europe, indicating the significance of our results and the strong signal-to-noise ratios at least over those continents. When applicable, more ensemble members for transient simulations can be used to further confirm the impacts of such variability. Furthermore, …**"

Greater details about the new ensemble runs and larger analysis periods of both the time-sliced and transient simulations are included below to respond to the reviewer's questions specific to these aspects.

*(2) It is clearly encouraging that "that the time-sliced experiments with single-year forcing looped for multiple years, give results very similar to the transient simulations, pointing to the robustness of LULCC impacts" (L767-769), but this very general statement would need to be expanded on and quantified.*

We have first rearranged our results such that the time-sliced simulations are presented first in Fig. 4 and transient (with two ensemble members) simulation presented in Fig. 7. We have included the comparison description between them in Sect. 3.3.3 of Results, by comparing Fig. 7 with Fig. 4. With additional ensemble members and the 30-years analysis period, we found again that the two sets of experiments share much similarities in ozone and meteorological changes, indicating the consistency of the LULCC signals. We have added a figure Fig. S6 in the supplementary materials to indicate their differences; we found that for most places the differences are statistically insignificant. The new Fig. 4, Fig. 7 and Fig. S6 are included below for your easy reference. We have modified Sect. 3.3.3 to read:

P33, Line 644-661: "**In the above sections, for a direct, parallel comparison with the Off-line configurations, we have used the time-sliced experiments with the present-day land cover in year 2000 and future land cover in year 2050. However, in reality the LULCC is transient with the land cover changing**

**gradually; therefore, transient runs in On-line mode with the land cover evolving from the present-day all the way to year 2065 are also conducted (On-line_45 and On-line_85, each with two ensemble members; see Table 1). Fig. 7 shows the changes in ozone and other variables from the transient simulations, using 2036 to 2065 as the 30-year averaging period to capture interannual variability. We find that changes in ozone, 2-m air temperature, and other factors controlling ozone are very similar between the transient and time-sliced runs (see also Table 2), with only statistically insignificant differences in different variables in most places (see Fig. S6 in the supplement). The consistent simulated results from the transient (Fig. 7) and time-sliced (Fig. 4) LULCC further reflect the robustness of the LULCC-induced signals at least over North America and Europe, which are strong enough to cause changes in meteorology and ozone pollution in places remote from LULCC, and indicate that** the atmospheric responses and biogeophysical effects are generally fast-responding at a quasi-steady state on timescales of years to decades with respect to the slow LULCC."

[Figure]

Figure 4. Simulated 2000-to-2050 changes in surface ozone, isoprene emission, dry deposition velocity and 2-m air temperature for the boreal summer averaged over the 30-year analysis window, under two future scenarios (RCP4.5 and RCP8.5) of LULCC. Regions with dots indicate changes that are significant at the 95% confidence level. These results are from the On-line runs (land forcing 2050

minus 2000) with dynamic meteorological responses to LULCC from time-sliced simulations On-line_45TS and On-line_85TS (Table 1).

[Figure]

Figure 7. Similar to Fig. 4 but these results are from the transient simulations On-line_45 and On-line85 (Table 1), averaged over the two ensemble members for each scenario.

[Figure]

Fig. S6. Differences between the time-sliced vs. transient simulated results, as shown in Fig. 4 and Fig. 6 of the main text. Statistically significant differences (>95% confidence) are indicated with dots.

*(3) Thus, in order to recommend publication I would need to be reasonably convinced the difference between their simulations is actually due to LULCC. To show this they need to show: simulations with land-cover changes are (1) significantly different from those without land-cover changes and (2) that this difference is due to the land-cover change itself. Without establishing both (1) and (2) we cannot be sure that the circulation changes causing the ozone differences are not due to atmospheric noise. Both Brovkin et al. (2013) and Lawrence et al. (2012) (referenced in the reviewed paper) examined the impact of land use changes on circulation. In each case they investigated the impacts of landuse from the difference between approximately 30-year runs. They both considered differences at the 95% level as significant. Brovkin et al. (2013) used multiple ensemble runs when available. The paper reviewed here uses the difference between 10-year averages and differences at the 90% level. Thus the statistical tests to distinguish between the simulations used in this paper are really quite weak and apparently not consistent with literature. To establish that the differences between the simulations are real:*

*-the paper should look at the 95% level consistent with other literature;*

*-they should also keep in mind that when considering hundreds of gridpoints some will appear consistent regardless (see Wilks et al., 2016; Bulletin of the American Meteorological Society 2016 vol: 97 (12) pp: 2263-2273)*

*-the paper should not discuss non-significant results as meaningful (see below)*

*Even if the differences between the simulations are real atmospheric 'noise ' can persist on the decadal timescale. Thus the authors need to address the fact the atmosphere can exhibit long timescale decadal changes which are due to low frequency variability, but not necessarily due to changes in LULCC. In other words the decadal simulations may be significantly different due to low frequency variability, but not due to changes in LULCC.*

To show results of "*(1) significantly different from those without land-cover changes and (2) that this difference is due to the land-cover change itself.*" suggested by the reviewer, we have additionally added a series of ensemble runs with slightly different initial conditions. The analysing period has also been revised to 30 years, for both the time-sliced and transient simulations. Details of the model settings are summarized in the revised Table 1 (included below). Due to limited time and computational resources, only two ensemble members of RCP4.5 LULC transient runs, and two ensemble members of RCP8.5 LULC transient runs have been added. For the time-sliced simulations, because the simulations are looped over the same year of land forcing (year 2000 and year 2050 for the present day and future, respectively), the 30 years of analysis could be considered as quasi-ensemble simulations with 30 members, and thus no additional ensemble runs are implemented. The time-sliced simulations are conducted in a way that has the most direct relevance for comparison with the Off-line simulations, except with a longer analysis period (30 years) to better capture potential interannual variability. We have therefore exchanged the order of presentation, such that the time-sliced results are presented first and foremost, and the transient results now come later.

| | Case Name | Land treatment | Meteorology | Simulated years | Model forcing |
|---|---|---|---|---|---|
| 1 | Off-line_CTL | Present-day (2000) land use and land cover (LULC) map | GEOS-5 reanalysis (2004-2017) | 14 years, the last 10 years for analysis | - Present-day (2000) well-mixed greenhouse gases and short-lived gases and aerosols, anthropogenic emissions; - Present-day (2000) monthly mean sea surface temperature and sea ice -All simulations use the SP mode in CLM - Isoprene emission is from MEGAN - Dry deposition velocity is based on Wesely (1989) updated by Val Martin et al. (2014) |
| 2 | Off-line_45 | 2050 RCP4.5 future LULC map as a time slice | Same as above | Same as above | |
| 3 | Off-line_85 | 2050 RCP8.5 future LULC map as a time slice | Same as above | Same as above | |
| 4 | On-line_CTL | Present-day (2000) LULC map | Simulated online | 60 years (looped over same year of forcing), the last 30 years for analysis | |
| 5 | On-line_45TS | 2050 RCP4.5 future LULC map as a time slice | Same as above | Same as above | |
| 6 | On-line_85TS | 2050 RCP8.5 future LULC map as a time slice | Same as above | Same as above | |
| 7, 8 | On-line_45[a] | 2000-2005 historical, 2006-2065 RCP4.5 transient LULC map | Same as above | 66 years (transient land forcing all the way), the last 30 years[c] for analysis | |
| 9, 10 | On-line_85[b] | 2000-2005 historical, 2006-2065 RCP8.5 transient LULC map | Same as above | Same as above | |

Table 1. List of model experiments. [a, b] Case 8 and 10 are in On-line_45 and On-line_85 are similar to Case 7 and 9, respectively, but with slightly different initial conditions to produce two ensemble members. [c] The analysis time period is from 2036 to 2065, centered around year 2050, as part of the transient land forcing.

We also revised the manuscript according to the reviewer's suggestions:

- The significant level has been revised from 90% level to 95% level, so as to be consistent with other literature.

- The paper (Wilks et al., 2016) pointed out that considering the collections of multiple statistical tests, often in the setting of individual tests at many spatial grid points, the outcomes of statistical tests is often overstated and overinterpreted. We have kept this in mind and first revised the significant level from 90% to 95% to increase enhance the significance of the scientific results. Since the areas with significant changes may be overinterpreted, the significant areas may be smaller than the shown areas. Therefore, discussions are only included for the areas with sufficiently large extent of significant changes (North America and Europe). Discussions on less significant results have been removed from the manuscript.

- We now analyse our model outputs based on 30-year average for both the time-sliced runs (looped over single-year forcing) and transient runs with two ensemble members.

Results from these experiments show that the general pattern of ozone changes is significant in North America and Europe, consistent between both LULCC transient runs and time-sliced runs (Fig. 4 and Fig. 7 above). The surface air temperature, as well as other meteorological conditions are altered in a way that can affect ozone through its impacts on isoprene and dry deposition. Compared with the previous results (10-year averages and one-member run for transient runs), the current results show some differences in ozone and meteorological changes and indeed are slightly less significant, indicating that low-frequency internal variability of the atmosphere does have some impact on the simulated signals. It corresponds well with the concern of the reviewer. However, the impact is limited, and significant and consistent changes can still be found in North America and Europe where substantial LULCC occurs.

Furthermore, we have indeed ran the two-member ensemble transient simulations all the way from the present day to 2080, and compared different 30-year analysing windows around 2050, to check to what extent low-frequency internal variability can affect the atmospheric responses to LULCC. See the figure below. The comparison shows very similar patterns of ozone changes regardless of which 30-year period is selected. It further indicates the limited effect of internal variability of the atmosphere on the strengths of LULCC-induced climate change signals as compared with the present day, and also indicates that LULCC can be strong enough to induce consistent changes of ozone. The period between 2036 to 2065 with the 2050 in the exact middle is currently presented in the revised manuscript.

[Figure]

**RCP4.5 projected TRANSIENT LULCC Surface O₃ (ppbv)**

Figure: Ozone changes in the LULCC two-member ensemble transient runs averaged using different 30-year analysis windows.

These results confirm the contribution of LULCC to climatic changes and the critical roles of LULCC in shaping surface ozone air quality through biogeophysical pathways. These revisions based on the combination of previous and new simulations and analysis periods have been adopted in the manuscript, and the related content has been revised accordingly.

We thank the reviewer for the valuable suggestions.

*(4) The simulations in Brovkin et al (2013) and Lawrence et al (2012) used 30-year averages (and in some cases ensemble simulations). This helps to distinguish the circulation differences in LULCC from atmospheric noise. The present paper used 10-year averages. One difference, however, is that the present paper uses fixed SSTs which might reduce the variability, but how much? I am not convinced 10 year averages are sufficient.*

We have expanded the analysis period to 30 years in both the time-sliced and transient simulations. Please see our responses to **comment # (3)** above.

*(5) This is a problem the authors need to address in a substantive way.*

*1) The best solution would be to run additional ensemble simulations. I believe these could be the time-slice experiments. My guess is that when you look at the 95% level you will need to run additional simulations anyhow. It is possible, to save cost, you could run the simulations*

*without the chemistry and just show the meteorological details are similar. It is not clear 55 year time-slice simulations are really necessary.*

We have now adopted two-member ensemble simulations for the transient LULCC experiments coupling both chemical and meteorological components. We have also switched the order of presentation: we now present the time-sliced simulations first to show a more direct comparison with the biogeochemical effects (Please see our detailed responses to **comment # (2) and (3)** above, as well as the revised Table 1 and Fig. 4 through Fig. 7).

The time-sliced simulations are revised to 60 years (still looping over the same year of forcing), with the last 30 years for analysis. Sixty years is not strictly necessary for the time-sliced simulations, only to the consistent with the integration time of the transient simulations. The spin-up period has also been carefully checked. The model can reach its equilibrium state after around 15 years when we use soil moisture at 10 cm as an indicator. If we consider soil moisture in deeper layers, additional years may be needed, thus, 30 years as spin-up period is chosen for the time-sliced and control simulations. More detail about the time-sliced and transient simulations are also included in Sect. 2.3:

P16, Line 331-350: "… The second and third experiments, On-line_45TS and On-line_85TS, are time-sliced simulations using 2050 land cover distribution following RCP4.5 and RCP8.5, respectively. **These two experiments are designed for direct, parallel comparison with the Off-line simulations, except with longer integration (60 years) and analysis (30 years) time to capture interannual climate variability. Because these multi-year simulations are looped over the same year of land cover forcing, they can be considered as a quasi-ensemble run and the multi-year average can be considered as the ensemble average. The fourth and fifth experiments, referred to as On-line_45 and On-line_85, are transient simulations performed continuously from year 2000 to 2065 using transient land cover maps projected for the RCP4.5 and RCP8.5 scenarios, respectively. These On-line transient simulations are repeated by a series of ensemble runs with slightly different initial conditions, with two ensemble members for each scenario. All the On-line experiments analysis is based on the last 30-year average and the ensemble average when modeled variables have attained a quasi-steady state. Comparison between the time-sliced and transient simulations helps us ascertain the strengths of LULCC-induced climate signals.**"

*2) It is possible the authors could make the case that the transient and timeslice simulations are similar enough that we can be reasonably sure the simulation differences are due to LULCC. However, I think the authors do need to make this case quantitatively. Where are the simulations the same, and where are they different? Again their difference should be distinguished using the 95th percent probability level. The authors mention these type of simulations are similar but do not address this in a quantitative manner. Moreover, the timeslice experiments are 55 years long. Does it really take the model that much time to reach equilibrium? It seems the authors could more effectively use the timescale experiments to establish the significance of the differences by looking at different 10 year intervals, for example.*

The comparison between the time-sliced and transient runs have been conducted and the implications for the strengths of the climate signals are also now discussed. Please see our responses to **comment # (2)** above.

*3) Hypothetically the authors could make dynamical arguments linking the changes in LULCC to the atmospheric changes. However, I am not convinced by the meteorological arguments in the paper. The meteorological differences between the simulations are of course self-consistent: the displacement of the jet-stream, the change in meridional temperature gradient, the changes in positions of anti-cyclones etc are consistent with each other, but this does not mean one can attributed these differences to LULCC. The albedo decreases seem to be related to the change in forest cover. However, neither of these seem very related to the changes in surface temperature or short-wave radiation. The northwards displacement of the jet occurs globally in two simulations with different land-cover changes and thus its relation to local changes in any one of the simulations is hard to discern. The authors make the point that the temperature increase occurs to the west of the region of LULCC but do not make a dynamic argument why this is so.*

In light of the new simulation results and in response to the reviewer's suggestions, we have substantially revised the mechanisms with stronger focuses on surface energy balance, boundary-layer meteorology, moisture transport and low-level atmospheric circulation changes. The jet stream analysis has been reduced and moved to the Conclusions and Discussion section in response to the reviewer's suggestions.

We have revised the manuscript as follows:

P27 Line 552-572: "For RCP4.5, North America is subjected to intensive regional changes in the land cover over the eastern US and southern Canada (Fig. 5d). Significant changes in surface ozone (Fig. 4a) and surface air temperature (Fig. 4d) are found over large continuous areas in North America, including both the regions with intensive LULCC and regions where LULCC is minimal. **Let us first focus on the forested regions with intensive LULCC (Fig. 5d), where reforestation results in a significant decrease in surface albedo (Fig. 5e). In the boreal and temperate mixed forests of southern Canada and northeastern US, such an albedo reduction results in a substantial enhancement in absorbed solar radiation (Fig. 5g). Typical of these forest types, the enhanced net radiation is in turn largely dissipated by higher sensible heat (Fig. 5h) instead of latent heat (Fig. 5i), resulting in a 0.5–1°C rise in average air temperature (Fig. 5j). This generates a warmer and drier boundary layer with suppressed precipitation (Fig. 5k), cloud cover (Fig. 5l), and soil moisture (Fig. 5o), constituting a feedback that likely further enhances net radiation. All these meteorological changes contribute to higher surface ozone concentrations (Fig. 5a) beyond the biogeochemical effects alone. In southern Canada, the drier conditions even help suppress dry deposition (Fig. 5c), further enhancing ozone there. These biogeophysical effects can be summarized by the cross-amplifying pathways in the blue box in Fig. 1. Furthermore, reduced wind speed (Fig. 5m) following enhanced roughness (as represented by vegetation height in Fig. 5f) may also reduce moisture transport to these forests, inducing a greater moisture divergence there (Fig. 5n).**

P28, Line 573-578: "**In contrast, in the subtropical broadleaf forests in the southeastern US, enhanced forest cover and albedo instead lead to greater**

moisture convergence from the Gulf of Mexico (Fig. 5n). This generates more favorable water conditions that not only dampen meteorological changes there but also promote dry deposition, leading to only slight changes in ozone. These can also be seen in the cross-counteracting pathways in the blue box of Fig. 1."

P30, Line 587-606: "Surface ozone also increases significantly over the locations where land use does not change **significantly, especially over the Midwest and Great Plains regions of north-central US (Figs. 5a and 5d). The ozone enhancement is found to correspond to the drier, warmer and sunnier conditions there that can be considered as "remote effects" of LULCC. Such conditions are associated with enhanced moisture divergence (Fig. 5n), which could be caused by the stronger convergence over the surrounding reforested regions that diverges moisture flow from the Great Plains, as well as reduced surface wind speed (Fig. 5m)** that can influence regional moisture transport to these regions (e.g., Sud et al., 1988; Xu et al., 2015). The vertically integrated moisture fluxes at present-day conditions are shown in Fig. S3a, illustrating that normally moisture transport from the Gulf of Mexico is deflected by the Rocky Mountains and toward the eastern and north-central US. **Due to reforestation, moisture transport is deflected further east and it generates an anomalous moisture flux divergence around the Midwest and Great Plains, resulting in drier conditions in these regions. The drier and warmer boundary layer are also reflected by the lower precipitation (Fig. 5k), cloud cover (Fig. 5l), soil moisture (Fig. 5o), latent heat (Fig. 5i), and the associated higher net radiation (Fig. 5g), sensible heat (Fig. 5h) and air temperature (Fig. 5j). The lower soil moisture can also reduce dry deposition there (Fig. 5c). All these changes can act together to enhance surface ozone over the north-central US as remote effects of LULCC elsewhere; these pathways can be summarized by the yellow box in Fig. 1.**"

P30-31, Line 609-621: "Substantial increases in surface ozone (Fig. 6a) and air temperature (Fig. 6j) are found in Europe due to the RCP4.5 LULCC scenario, whereby substantial reforestation occurs **over in the boreal and temperate mixed forests** in the European continental regions (Fig. 6d), modifying surface energy balance significantly**. Over the regions with intensive LULCC, the biogeophysical pathways shaping boundary-layer meteorology and ozone are largely similar to southern Canada and northeastern US, where the forest types are similar (see blue box in Fig. 1). In brief, reduced albedo (Fig. 6e) leads to enhanced net radiation (Fig. 6g) and sensible heat (Fig. 6h), raising surface air temperature over a large area by 0.4–1.2°C (Fig. 6j), and constituting a hydrometeorological feedback that reduces precipitation (Fig. 6k), cloud cover (Fig. 6l), and soil moisture (Fig. 6o). These changes generate warmer, drier and sunnier conditions over the forests that favor higher ozone levels. Reforestation also decreases surface wind speed (Fig. 6m) and moisture transport at the near-surface level.**"

P31, Line 622-629**: "The increases in surface ozone are also found to extend westward and southward beyond the regions with intensive LULCC, likely reflecting remote effects (Fig. 6a). The lower-level wind patterns at 850 hPa under present-day conditions are shown in Fig. S3b, showing that reforested regions are originally on the southerly branch (eastern part) of the Azores High anticyclone. Circulation changes in response to reforestation appears to enable the anticyclonic system to extend eastward, allowing sunny and warm conditions typical of the Azores High to prevail over much of western Europe and parts of North Africa, and enhancing surface ozone there.**"

Fig. 1 has also been substantially revised as follows:

[Figure]

Figure 1. Schematic diagram showing the biogeochemical and biogeophysical effects of any changes in the forest cover resulting from land use and land cover change (LULCC) on surface ozone. Red arrows indicate the biogeochemical pathways and grey arrows indicate the biogeophysical effects via changes in the overlying meteorological environment. The sign associated with each arrow indicates the correlation between the two variables; the sign of the overall effect (positive or negative) of a given pathway is the product of all the signs along the pathway. We here focus on processes initiated on the land surface by LULCC, and the corresponding responses in local near-surface atmosphere (blue box) and remote near-surface atmosphere (yellow box).

The abstract has also been revised accordingly for the parts on the biogeophysical effects:

"… reflecting the importance of biogeophysical effects on ozone changes. **In boreal and temperate mixed forests with intensive reforestation, enhanced net radiation and sensible heat induce a cascade of hydrometeorological feedbacks that generate warmer and drier conditions favorable for higher ozone levels. In contrast, reforestation in subtropical broadleaf forests has minimal impacts on boundary-layer meteorology and ozone air quality.** Furthermore, significant ozone changes are also found in regions with only modest LULCC, which can only be explained by "remote" biogeophysical effects. **A likely mechanism is that**

**reforestation induces a circulation response, leading to reduced moisture transport and ultimately warmer and drier conditions in the surrounding regions with limited LULCC.** We conclude that the biogeophysical effects of LULCC are important pathways **through which LULCC influences ozone air quality** both locally and in remote regions even without significant LULCC. Overlooking **the effects of hydrometeorological changes on ozone air quality** may cause underestimation of the impacts of LULCC on ozone pollution."

The Conclusions and Discussion section has also been revised:

P36, Line 702-712: "The mechanisms behind **hydrometeorological** responses to LULCC are summarized in Fig. 1. **In brief, first, surface properties and processes (e.g., surface albedo and evapotranspiration) are altered, leading to changes in the surface energy balance. In boreal and temperate mixed forests, the albedo effect dominates, leading to higher net radiation, sensible heat, and surface temperature, but reduced precipitation, cloud cover and soil moisture.** These local changes can also induce a regional circulation response, in particular the formation of anomalous **moisture divergence and corresponding warmer and drier conditions over the surrounding regions even with limited LULCC. In subtropical broadleaf forests, however, both the albedo and evapotranspiration effects are important and they tend to offset each other, leading to minimal hydrometeorological changes.**"

P36-37, Line 713-724: "**In our analysis of LULCC-induced hydrometeorological changes, we have focused on the surface and the overlying boundary layer. Many studies have found that LULCC-induced surface changes can propagate to upper levels as high as 200 hPa (e.g., Chase et al., 2000; Swann et al., 2012; Medvigy, et al., 2013; Xu et al., 2015; Jia et al., 2019). In our study, significant meteorological changes can be detected at the upper levels up to 200 hPa due to LULCC (not shown), which can lead to circulation changes, storm track displacement, and anomalous subsidence especially at midlatitudes, likely constituting feedbacks on precipitation, moisture transport, and temperature. However, we find no clear conclusions as to whether these upper-level changes and feedbacks could have sufficient influence on ozone-relevant hydrometeorological conditions beyond that can be explained by boundary-layer dynamics alone.**"

*(6) The authors should refrain from discussion non-significant signals. This is particularly the case when discussing the signal over Europe. From figure 7 the ozone changes are only significant over the ocean (at the 90th percentile level) and the temperature changes are not significant anywhere over Europe. Most of the changes discussed in this section are not significant at the 90th percentile. The discussion of changes in Europe should probably be dropped.*

We have removed results with less significant changes, and the significant level has been improved to 95%.

*Minor Comments.*

*1. India. The precipitation increase appears to be displaced southward, not northward as stated in the text. I do not see evidence for an anticyclone in the figures, nor the consistency between the significant change in rainfall which seems to occur in the south of the domain and a displace cyclone.*

> According to the additional ensemble runs, the changes in surface ozone are significant but are limited to only a narrow band of areas over the Himalaya. Therefore, results on this part have been removed.

*2. Figure 1 is a nice figure, but in some ways is misleading. The thermal wind du/dz ~ dT/dy, and thus the jet is not related to dT/dy at a particular level as implied in Figure 1. Moreover, a displacement of the jet-stream does not lead necessarily lead to an anomalous high more than an anomalous low. It.is well established that changes in LULCC lead to upper level changes but the precise nature of these changes are probably quite complex.*

> We have revised the Fig. 1 thoroughly, please refer to the response to **comments # (5)** above. In particular, we have ascribed most of the meteorological and ozone changes to changes in the surface energy balance, hydrometeorology and boundary-layer dynamics, instead of upper-level changes.

*3. The significance in the figures is still difficult to read. Some authors only color the parts of the diagrams that are significant, but there are other solutions.*

> Figures have been revised: significant areas are circled to show clearly the significantly changed regions.

*4. L254: upper and mid-troposphere ozone observations?*

> P12, L269-275, revised as "In general, CAM-Chem can reasonably replicate observed values at individual sites …, and mid- and upper-tropospheric  **distribution derived from a compilation of ozone measurements** (Lamarque et al., 2010; **Cooper et al., 2010**) albeit with a general overestimation. The performance is comparable to other global and regional models (Lapina et al., 2014; Parrish et al., 2014)."

*5. L277: "emissions": anthropogenic emissions?*

> P13, L297, revised as "In this study, **anthropogenic** emissions are held constant at the present-day level for all runs, …".

*6. Table 1: the last column "Other settings" is confusing as it is not clear what simulations this applies to.*

> We now revised the Table 1 in the manuscript. Please see our responses to **comment # (3)**.

*7. L332: "integrated": do you mean combined?*

> We have checked through the manuscript, and revised the word mentioned by the reviewer accordingly.

P17, L357: "(3) the **combined**  effects induced by LULCC on surface ozone and its precursors and dry deposition.".

*8. L439: "can be reduced". This is not clear.*

Revised as P23, L465: "Recent studies found that these peroxides can be rapidly photolyzed, making them at best a temporary $HO_x$ reservoir (e.g., Thornton et al., 2002; Kubistin et al., 2010)."

**Abstract**

Surface ozone ($O_3$) is an important air pollutant and greenhouse gas. Land use and land cover is one of the critical factors influencing ozone, in addition to anthropogenic emissions and climate. Land use and land cover change (LULCC) can on the one hand affect ozone "biogeochemically", i.e., via dry deposition and biogenic emissions of volatile organic compounds (VOCs). LULCC can on the other hand alter regional- to large-scale climate through modifying albedo and, evapotranspiration, which can lead to changes in surface temperature, hydrometeorology and atmospheric circulation that can ultimately impact ozone "biogeophysically" over local and remote areas. Such biogeophysical effects of LULCC on ozone are largely understudied. This study investigates the individual and combined biogeophysical and biogeochemical effects of LULCC on ozone, and explicitly examines the critical pathway for how LULCC impacts ozone pollution. A global coupled atmosphere-chemistry-land model is driven by projected LULCC from the present day (2000) to future (2050) under RCP4.5 and RCP8.5 scenarios, focusing on the boreal summer. Results reveal that when considering biogeochemical effects only, surface ozone is predicted to have slight changes by up to 2 ppbv maximum in some areas due to LULCC. It is primarily driven by changes in isoprene emission and dry deposition counteracting each other in shaping ozone. In contrast, when considering the  combined effect of LULCC, ozone is more substantially altered by up to  5 ppbv over several regions in North America and Europe under RCP4.5, reflecting the importance of biogeophysical effects on ozone changes. In boreal and temperate mixed forests with intensive reforestation, enhanced net radiation and sensible heat induce a cascade of hydrometeorological feedbacks that ultimately generate warmer and drier conditions favorable for higher ozone levels. In contrast, reforestation in subtropical broadleaf forests has minimal impacts on boundary-layer meteorology and ozone air quality. Furthermore, significant ozone changes are also found in regions with only modest LULCC, which can only be explained by "remote" the biogeophysical effects. A likely mechanism is that enhanced convergence in the reforested regions induces a circulation response, leading to enhanced divergence, 
[revised manuscript text omitted]
 together to promote enhance warming higher surface ozone over the north-central central-western US as remote effects of LULCC

region elsewher; these pathways can be summarized by the yellow box in Fig. 1.

For the RCP8.5 run, surface ozone is also enhanced in North America (Fig. 4e) and is again well correlated with near-surface warming (Fig. 4h, Fig. S2d). However, the ozone concentration increase is smaller than that in RCP4.5, presumably due to weaker LULCC.

[Figure]

Figure 6. Present-day conditions and changes in zonal wind at 200 hPa and geopotential height at 500 hPa during the boreal summer. Changes due to RCP4.5 projected LULCC change are in middle-bottom left panel, while RCP8.5 in middle-bottom right panel. Regions with dots indicate changes that are significant at the 90% confidence level.

**3.3.2 Europe for under RCP4.5 reforestation and RCP8.5**

Along coastal areas of Europe, Ssubstantial increases in surface ozone (Figs. 64a, e) and near-surface air temperature (Figs. 6i4d, h) are found in Europe due to the RCP4.5 and RCP8.5 LULCC. For RCP4.5scenario, whereby substantial reforestation occurs over in the boreal and temperate mixed forests in the European continental regions (Fig. 7b6d), which modifyingies regional surface energy balance significantlyand atmospheric circulation. Over the regions with intensive LULCC, the biogeophysical pathways shaping boundary-layer meteorology and ozone are largely similar to southern Canada and northeastern US, where the forest types are similar (see blue box in Fig. 1). In brief, reduced  albedo (Fig.

6e) leads to enhanced net radiation (Fig. 6g) and sensible heat (Fig. 6h), raising surface air temperature over a large area by 0.4–1.2 K (Fig. 6j)

, and constituting a hydrometeorological feedback that reduces precipitation (Fig. 6k), cloud cover (Fig. 6l), and soil moisture (Fig. 6o). These changes generate warmer, drier and sunnier conditions over the forests that favor higher ozone levels.

Reforestation also decreases surface wind speed (Fig. 6m) and moisture transport at the near-surface level, resulting in greater moisture divergence over the forests (Fig. 6n).

The increases in surface ozone are also found to extend westward and southward beyond the regions with intensive LULCC, likely reflecting remote effects (Fig. 6a). The lower-level wind patterns at 850 hPa under present-day conditions are shown in Fig. S3b, showing that reforested regions are originally on the southerly branch (eastern part) of the Azores High anticyclone. Stronger dry convection over the reforested regions appears to enable the anticyclonic system to extend eastward, allowing sunny and warm conditions typical of the Azores High to prevail over much of western Europe and parts of North Africa, and enhancing surface ozone there.

[Figure]

[Figure]

Figure 76. Similar to Fig.ure 5 but for Europe in under RCP4.5.

Again, this is likely due to a similar mechanism in which the northward-migrating storm track (Fig. 7f) and westerlies (Fig. 6c) are found at about 55–60°N; modified storm tracks and the anomalous high are acting in concert, leading to more subsidence in the European region that experiences increased surface net solar radiation (Fig. 7d), thus surface warming (Fig. 7e). For RCP8.5, reforestation occurs over limited areas of

Europe (Figs. 2l); similar changes in the local climate and surface ozone are found, albeit with a relatively weak amplitude compared with their RCP8.5 counterparts.

3.3.3 India and China for RCP4.5 and RCP8.5

For RCP4.5, extensive reforestation occurs in northeastern and southwestern India (Fig. 8b). There is also a significant increase of surface ozone over northern India (Fig. 4a), collocated with warming (Fig. 4d). Again, temperature increase tends to occur west of the LULCC (Fig. 8e). The LULCC-induced lower albedo (Fig. 8c) and higher net surface solar radiation (Fig. 8d) cause more energy to be absorbed by the land surface at high elevations and warm the overlying air accordingly. Again, the rainbelt is displaced northward, likely reflecting perturbed synoptic-scale activities in the region. Consistent with the former feature, the mid-tropospheric anomalous flow is characterized by an anticyclone between 20–30°N, suppressing rainfall therein (Figs. 8f, 8g). The anomalous anticyclone in turn can lead to more surface net radiation in northern India as a remote effect (Fig. 8d). Thus, in northern India there is significant surface warming (Fig. 8e) and enhanced surface ozone.

[Figure]

Figure 8. Similar to Figure 5 but for India in RCP4.5.

Finally, in China extensive deforestation occurs for RCP4.5 (Fig. 2f). Surface ozone shows a slightly decrease (Fig. 4a) that could be caused by biogeochemical effects associated with LULCC instead of biogeophysical effects. This region is characterized by a temperate climate, medium isoprene emission from temperate trees (Fig. 3a) and high anthropogenic $NO_x$ emissions. Changes from temperate trees to croplands further decrease isoprene emission and lead to significant ozone decreases, which largely offsets the effects of reduced dry deposition velocity (Fig. 4b). For

RCP8.5, little change in surface ozone or temperature has been found in either country.

Overall, we find that biogeophysical effects can have strong impacts on surface ozone through modifying local and remote meteorological conditions such as surface warming, drying and circulation anomalies initiated by local LULCC (Fig. 1).

Our results of temperature changes are consistent with the previous study of Swann et al. (2012) that illustrated the local and remote climate effects of the northern midlatitude reforestation. They conducted a model experiment with extreme afforestation, and found substantial warming in North America and Europe. In addition, Govindasamy and Caldeira (2001) and Unger (2014) also found surface cooling due to deforestation.

3.3.4 3 Transient time-sliced experiments versus time-slice transient experiments

In the above sections, for a direct, parallel comparison with the Off-line configurations, we have use the time-sliced experiments for with the present-day land cover conditions in year 2000 and future conditions 
[revised manuscript text omitted]